# A segregated cortical stream for retinal direction selectivity

Rune Rasmussen[1,2], Akihiro Matsumoto [1,2], Monica Dahlstrup Sietam [1] & Keisuke Yonehara [1 ✉]

Visual features extracted by retinal circuits are streamed into higher visual areas (HVAs) after being processed along the visual hierarchy. However, how specialized neuronal representations of HVAs are built, based on retinal output channels, remained unclear. Here, we addressed this question by determining the effects of genetically disrupting retinal direction selectivity on motion-evoked responses in visual stages from the retina to HVAs in mice. Direction-selective (DS) cells in the rostrolateral (RL) area that prefer higher temporal frequencies, and that change direction tuning bias as the temporal frequency of a stimulus increases, are selectively reduced upon retinal manipulation. DS cells in the primary visual cortex projecting to area RL, but not to the posteromedial area, were similarly affected. Therefore, the specific connectivity of cortico-cortical projection neurons routes feedforward signaling originating from retinal DS cells preferentially to area RL. We thus identify a cortical processing stream for motion computed in the retina.

[1] Danish Research Institute of Translational Neuroscience—DANDRITE, Nordic-EMBL Partnership for Molecular Medicine, Department of Biomedicine, Aarhus University, 8000 Aarhus C, Denmark. [2]These authors contributed equally: Rune Rasmussen, Akihiro Matsumoto. ✉email: keisuke.yonehara@dandrite.au.dk

The mammalian visual system analyzes the external world through a set of distinct spatio-temporal channels[1]. The mouse retina contains >40 distinct types of ganglion cells, each encoding a discrete set of visual features such as color, luminance, edges, and motion direction[2]. The general consensus is that central visual areas combine signaling originating from different ganglion-cell types[3], and the output from each ganglion-cell type diverges into multiple central visual areas[4], embodying a feature-combinatorial system[5].

The mouse visual cortex includes up to 16 retinotopically organized higher visual areas (HVAs), varying in preferences for temporal and spatial frequency, motion speed, color, and visual-field coverage[6–10]: these are categorized into a dorsal and ventral stream dichotomy[7,11,12]. In rodents, functionally segregated streams are already observed in the cortico-cortical projection neurons of the primary visual cortex (V1)[13,14]. However, it is not yet understood how signaling originating from individual retinal channels influences the activity of HVAs and, hence, how HVA-specific properties and distinct visual streams emerge[15,16]. One possibility is that each retinal channel influences all HVAs to a similar extent. Alternatively, in an extreme scenario, the impact of each retinal channel may be localized to a single HVA. If so, it will be critical to determine how V1 cortico-cortical projection neurons integrate inputs originating from individual retinal ganglion-cell types.

One fundamental task of the visual system is to detect the direction and speed of visual motion to analyze object motion or optic flow. The direction and speed of visual motion are first encoded by retinal direction-selective (DS) cells, preferentially responding to visual stimuli moving in a particular direction[17,18]. Retinal ON-OFF DS cells include four subtypes: each prefers one of four cardinal directions[19–21]. In mice, the shell region of the dorsal lateral geniculate nucleus (dLGN) relays signals from retinal horizontal ON-OFF DS cells to superficial layers of V1 (refs. [22–24]), and genetic knockout of retinal horizontal direction selectivity reduces a posterior motion preference in V1 layer (L) 2/3 DS cells[25]. However, how direction selectivity originating in the retina is processed beyond V1 remains unexplored. It could be broadcasted to all HVAs equally or, alternatively, it may be preferentially transmitted to specific HVAs. In addition, direction selectivity is also computed de novo at the thalamocortical synapses in L4 of V1 in mice[26], but it remains unknown whether retina-dependent and -independent direction selectivity mechanisms are combined or stay segregated in HVAs.

To probe these questions, we used *Frmd7* mutant mice (*Frmd7*[tm]) to disrupt horizontal direction selectivity in the retina[25,27,28], and transgenic mice expressing diphtheria toxin receptors in starburst amacrine cells (*ChAT-Cre × LSL-DTR*) to genetically ablate the cells, leading to the loss of retinal direction selectivity[25,29]. We tested the effect of these manipulations on visual motion processing in the retina, thalamic axons, V1, and HVAs in anaesthetized and awake mice. We show that the preference of the rostrolateral (RL) area for higher temporal frequencies (TFs) in the posterior direction is the major response feature affected by the alteration of retinal motion computations. We determine a functional pathway that links retinal horizontal DS cells to area RL while bypassing L4 of V1: dLGN → V1 L2/3 → RL. Importantly, V1 L2/3 neurons that project to area RL, but not the posteromedial (PM) area, were affected by the disruption of retinal horizontal direction selectivity, indicating a segregated V1 circuitry that routes retinal DS signaling preferentially to area RL. Our results indicate there is a cortical space for retinal direction selectivity and a distinct pathway that enables specialized response properties in HVAs.

## Results

**Mapping the sensitivity of cortical areas.** We mapped the visual cortex organization in anesthetized control and *Frmd7*[tm] mice using intrinsic signal optical imaging (ISOI; Fig. 1a). We generated visual-field sign maps (Fig. 1a) from horizontal and vertical retinotopic maps, and reliably identified six visual areas: V1, lateromedial (LM), anterolateral (AL), RL, anteromedial (AM), and PM (Fig. 1a). It is worth noting that the RL area we identified may possibly include the anterior HVA (area A), which has been identified previously[6,7]. We found no differences in visual cortical organization or size proportions between control and *Frmd7*[tm] mice (Supplementary Fig. 1). To test the contribution of retinal direction selectivity to motion responses in visual areas, we measured evoked intrinsic signal activity levels[12] in response to gratings drifting in the cardinal directions at a TF of 0.3, 0.75, 1.2, or 1.8 Hz with a fixed spatial frequency of 0.03 cycles/°. In areas V1 and RL, we found significantly decreased responses to horizontal motion in *Frmd7*[tm] mice at multiple TFs (Fig. 1b). In control mice, these areas responded strongly to horizontal motion moving at higher TFs (Fig. 1b). In areas LM and AM, only posterior responses at 1.2 Hz were significantly decreased and increased, respectively, in *Frmd7*[tm] mice (Fig. 1b). These findings suggest that retinal horizontal direction selectivity contributes to motion responses in a subset of visual cortical areas. In particular, the higher-TF preference of areas V1 and RL for horizontal motion was dominantly impaired in mice with disrupted retinal horizontal DS signaling.

**RL DS cells rely on retinal direction selectivity.** To elucidate the cellular underpinnings of the ISOI results, we used in-vivo two-photon calcium imaging from anesthetized mice (Fig. 1c and Supplementary Fig. 2). We focused on the areas RL and PM because RL was notably affected in *Frmd7*[tm] mice, whereas PM was unaffected (Fig. 1b). We imaged neurons in L2/3 using the virally transduced GCaMP6f, and stimuli consisted of gratings drifting in 12 directions at TFs of 0.3, 0.75, 1.2, or 1.8 Hz with a fixed spatial frequency of 0.03 cycles/° (Fig. 1d–f). RL DS cells (direction selectivity index [DSI] > 0.3) in control mice preferred higher TFs, particularly in the posterior direction (Fig. 1g, h), whereas PM DS cells preferred low TFs at a similar level in all directions (Fig. 1g, h). In *Frmd7*[tm] mice, RL DS cells preferred lower TFs (Fig. 1g, h), whereas the preference of PM DS cells was unchanged (Fig. 1g, h). We found no differences in vertical motion responses in area RL or PM between control and *Frmd7*[tm] mice (Fig. 1h). In both control and *Frmd7*[tm] mice, RL DS cells developed a posterior bias in the distribution of preferred directions as the TF increased from 0.3 to 1.2 Hz ($P \geq 0.05$ and $P < 0.01$, Rayleigh test; Fig. 1i): however, a notably smaller fraction of DS cells in RL of *Frmd7*[tm] mice preferred posterior motion (29.3%) compared to control mice (48.1%) at 1.2 Hz (Fig. 1j). Preferred directions of PM DS cells showed a posterior bias at 0.3 and 1.2 Hz ($P < 0.001$, Rayleigh test; Fig. 1i) in both control and *Frmd7*[tm] mice (Fig. 1i, j).

Anesthesia is known to influence, for example, cortical dynamics and synaptic excitation and inhibition[30,31]. Thus, to validate our findings, we repeated the experiment in awake, quietly resting mice (Supplementary Fig. 3). Overall, these experiments confirmed that the key findings in anesthetized mice (Fig. 1f–j) were preserved in awake mice. RL DS cells in awake control mice preferred motion moving along the horizontal axis at higher TFs, while PM DS cells preferred motion at low TFs equally across all directions (Supplementary Fig. 3c, d). In contrast, RL DS cells in awake *Frmd7*[tm] mice preferred low TFs, whereas the preference of PM DS cells was unchanged (Supplementary Fig. 3c, d). Importantly, as the

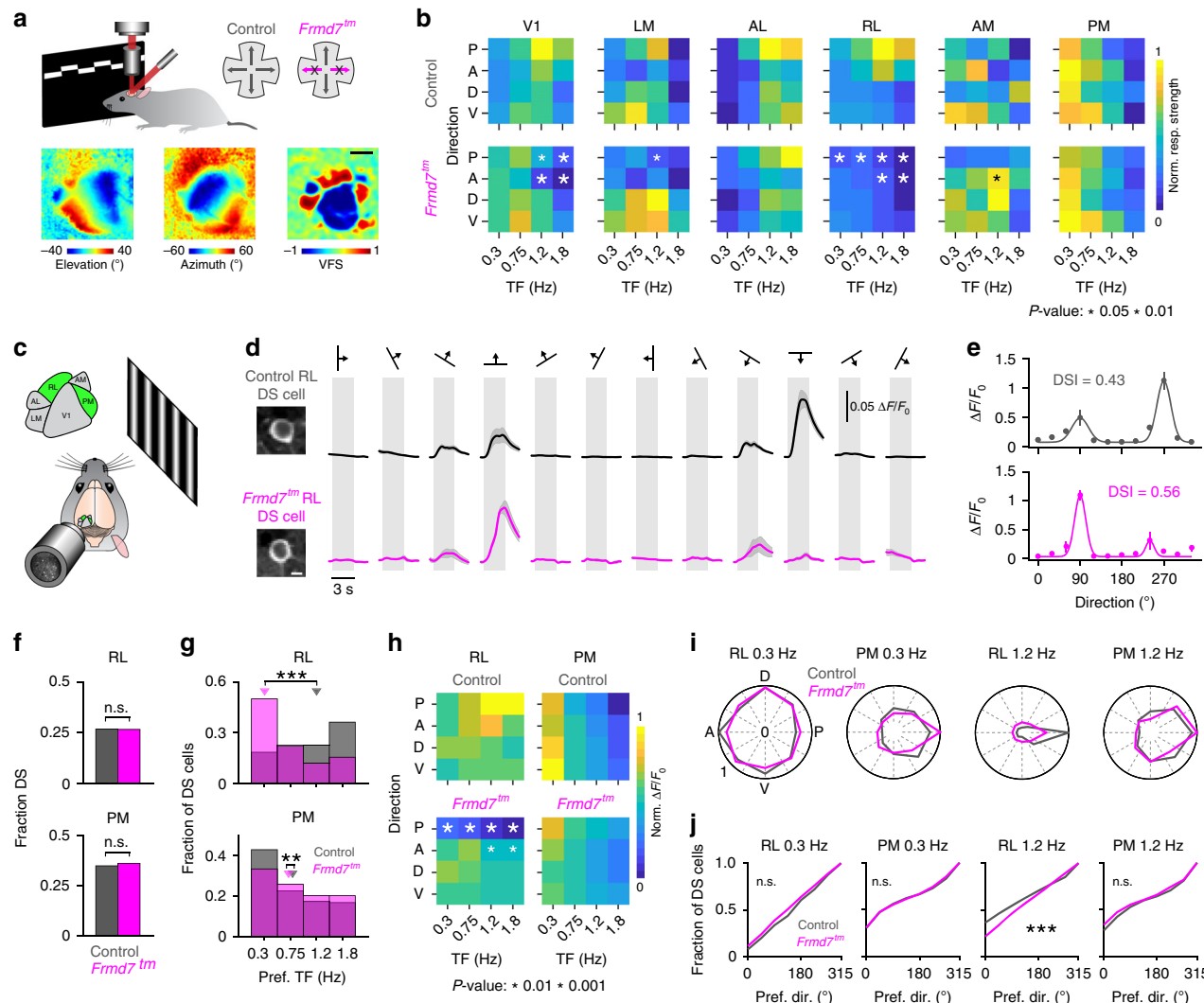

**Fig. 1 Posterior motion preference of RL neurons at higher TFs depends on retinal horizontal direction selectivity. a** Upper: ISOI in control and *Frmd7tm* mice. Lower: Example vertical and horizontal retinotopic maps and the computed visual-field sign map. Scale bar, 1 mm. **b** Response strength as a function of motion direction (anterior [A], posterior [P], dorsal [D], and ventral [V]) and TF (five mice per genetic group). White and black asterisks: significantly decreased and increased responses, respectively, two-sided Mann–Whitney *U*-test. **c** Two-photon calcium imaging from L2/3 in areas RL and PM of control (1452 and 1098 DS cells, respectively; ten mice) and *Frmd7tm* mice (1387 and 1217 DS cells, respectively; 11 mice). **d** Example control and *Frmd7tm* RL and PM neurons expressing GCaMP6f (scale bar, 5 μm) and trial-averaged fluorescence ($\Delta F/F_0$) time courses for the same neurons. Shading indicates SEM. **e** Tuning curves for neurons shown in **d**. Error bars are SEM; solid line is Gaussian fit. **f** Fraction of DS cells in RL and PM (two-sided $\chi^2$ test with Yates correction). **g** Preferred TF for DS cells in RL (two-sided Mann–Whitney *U*-test) and PM (two-sided Mann–Whitney *U*-test). Triangles show medians. **h** Response amplitude as a function of motion direction and TF for RL and PM DS cells. White asterisks: significantly decreased response amplitude in *Frmd7tm* mice, two-sided Mann–Whitney *U*-test. **i** Fractional distributions of preferred motion directions for RL and PM DS cells at 0.3 and 1.2 Hz; fractions are normalized to the largest fraction across genetic groups. **j** Distributions of preferred direction at 0.3 and 1.2 Hz in RL and PM (two-sided Kolmogorov–Smirnov test). **P < 0.01, ***P < 0.001, n.s., not significant, in **f**, **g**, and **j**. Source data are provided as a Source Data file.

TF increased from 0.3 to 1.2 Hz, RL but not PM DS cells from awake control mice developed a strong bias for posterior motion, and this bias was significantly impaired in *Frmd7tm* mice (Supplementary Fig. 3e, f). These results suggest the following: First, increased response amplitude at higher TFs is correlated with a gain of bias towards posterior motion in RL DS cells. Second, these TF-dependent changes in response amplitude and directional preference in area RL depend on retinal horizontal direction selectivity. Third, the effect of disrupting retinal direction selectivity appears to be specific to certain HVAs.

**RL DS cells are impaired by retinal starburst cell ablation.** The effect of *Frmd7* mutation on starburst cells is chronic from birth;

potentially triggering plasticity-related changes in the downstream visual pathways. Thus, we next genetically ablated retinal starburst amacrine cells in adult mice to acutely abolish retinal direction selectivity. For this we used *ChAT-Cre × LSL-DTR* mice in which diphtheria toxin receptors are selectively expressed in starburst amacrine cells[25,29]. Intravitreal injection of diphtheria toxin into these mice led to the selective ablation of starburst amacrine cells (Supplementary Fig. 4; referred to as 'starburst-ablated mice') and a loss of optomotor responses (OMR; Supplementary Fig. 4). We imaged neurons in L2/3 of areas RL and PM in awake, quietly resting mice (Supplementary Fig. 5). We found no differences in the fraction of DS cells between control and starburst-ablated mice in area RL or PM (Supplementary Fig. 5b). In starburst-ablated mice the distribution of TF preference was shifted toward slower

TFs for RL DS cells compared to control mice, whereas we found no difference for PM DS cells (Supplementary Fig. 5c). Similar to what we observed in *Frmd7tm* mice, RL DS cells from starburst-ablated mice showed a lack of horizontal motion response preference at higher TFs, in contrast to RL DS cells from control mice (Supplementary Fig. 5d). Notably, the posterior motion bias at 1.2 Hz was significantly impaired in RL DS cells from starburst-ablated mice (19.4% of DS cells) compared to control mice (29% of DS cells), whereas we found no difference for PM DS cells (Supplementary Fig. 5e). Thus, the key findings from *Frmd7tm* mice were largely supported by results from starburst-ablated mice.

**Distinct RL neurons rely on retinal motion computation**. To test whether there is a trend in affected response properties in *Frmd7tm* mice, we performed decomposition and segmentation on the datasets from areas RL and PM. First, we composed a TF-dependent response matrix for RL L2/3 DS cells, pooled from control and *Frmd7tm* mice (Fig. 2a). Next, we used principal component analysis (PCA) to decompose the response matrix into two dimensions (Fig. 2b). Noticeably, the PCA distribution showed a clear distribution trend depending on the TF preference: neurons sharing the same TF preference tended to be locally clustered (Fig. 2c). We then determined the fraction of RL neurons in local regions of the PCA distribution by superimposing 8 × 8 grids (Fig. 2d and Supplementary Fig. 6). Statistical comparisons of fractions between control and *Frmd7tm* mice revealed grids where the fraction of neurons was decreased, increased, or unchanged in *Frmd7tm* mice (Fig. 2e). Next, we probed the functional characteristics of these three groups. Notably, RL neurons that were decreased in *Frmd7tm* mice showed a prominent response amplitude increase as the TF increased ($P < 0.001$ for 0.3 versus 1.2 Hz for control and *Frmd7tm* mice, Wilcoxon signed-rank; Fig. 2f). In contrast, neurons that were increased or unchanged in *Frmd7tm* mice showed a TF-dependent decrease in response ($P < 0.05$ for 0.3 versus 1.2 Hz for control and *Frmd7tm* mice, Wilcoxon signed-rank; Fig. 2f). Neurons that were decreased in *Frmd7tm* mice showed a clear shift in direction tuning: at 0.3 Hz neurons uniformly encoded all directions while at 1.2 Hz they developed a preference for posterior motion together with an increase in DSI ($P < 0.001$ for 0.3 versus 1.2 Hz for control and *Frmd7tm* mice, Wilcoxon signed-rank; Fig. 2g). In contrast, neurons that were increased or unchanged in *Frmd7tm* mice showed no obvious direction tuning bias at either low or high TFs, but did decrease their DSI at the higher TF ($P < 0.001$ for 0.3 versus 1.2 Hz for control and *Frmd7tm* mice, Wilcoxon signed-rank; Fig. 2g).

We repeated the above analysis for PM L2/3 DS cells (Fig. 2h). Similar to in area RL, neurons from area PM showed a TF-dependent PCA distribution trend (Fig. 2i, j). We found no grids where the fraction of neurons was significantly different between control and *Frmd7tm* mice (Fig. 2k, l). PM neurons decreased their response amplitude as a function of increasing TF (Fig. 2m), with no clear change in direction tuning bias or DSI ($P \geq 0.05$ for 0.3 versus 1.2 Hz for control and *Frmd7tm* mice, Wilcoxon signed-rank; Fig. 2n). Together, these data indicate that the coupling of a high-TF motion preference and the development of posterior motion bias at high TFs is the major response feature that depends on retinal horizontal direction selectivity. Thus, the retinal contribution to motion processing in HVAs appears to be specific not only to area, but also to ensemble.

**Projection-specific impairment of V1 neurons**. Previous work has shown that V1 neurons provide target-specific input to HVAs[13,14]. Here, we investigated whether the sensitivity to disruption of retinal horizontal direction selectivity is routed by V1 DS cells that project to areas RL and PM. Alternatively, such sensitivity may originate from inputs from other HVAs or higher-order thalamic areas. To probe this, we labeled RL- and PM-projecting V1 L2/3 neurons by injecting rAAV2-retro expressing GCaMP6m into either area PM or RL (Supplementary Fig. 7) and imaged these neurons and target-unspecific V1 L2/3 neurons in anaesthetized mice (Fig. 3a–c). The density of GCaMP6-labeled projection neurons did not differ significantly between genetic conditions (Supplementary Fig. 7e). Notably, the fraction of RL-projecting neurons that showed direction selectivity was reduced from 60% in control mice to 34% in *Frmd7tm* mice, whereas the fraction of DS cells among PM-projecting neurons was not altered in *Frmd7tm* mice (Fig. 3d). RL-projecting and target-unspecific DS cells from control mice preferred higher TFs, particularly in horizontal directions (Fig. 3e, f). In *Frmd7tm* mice, RL-projecting and target-unspecific DS cells preferred the lowest TF, showing lower responses than control mice to horizontal motion at higher TFs (Fig. 3e, f). PM-projecting DS cells preferred low TFs for all directions in control mice, and this was the same in *Frmd7tm* mice (Fig. 3e, f). Consistent with previous work[25], target-unspecific neurons developed a posterior bias in the distribution of preferred directions as the TF increased ($P \geq 0.05$ and $P < 0.05$ for 0.3 and 1.2 Hz, respectively, Rayleigh test) with significantly less bias in *Frmd7tm* than in control mice at 1.2 Hz (Fig. 3g, h). PM-projecting DS cells from both control and *Frmd7tm* mice developed a posterior bias as the TF increased from 0.3 to 1.2 Hz ($P \geq 0.05$ and $P < 0.001$, Rayleigh test; Fig. 3g, h). RL-projecting DS cells from control mice developed a posterior bias ($P \geq 0.05$ and $P < 0.001$ for 0.3 and 1.2 Hz, respectively, Rayleigh test), but this did not occur in *Frmd7tm* mice (Fig. 3g, h). These data suggest that the distinct sensitivity of areas RL and PM to disruption of retinal horizontal direction selectivity is already found in V1 L2/3 DS cells that project to these areas.

**Distinct V1 neurons rely on retinal motion computation**. Next, we correlated the response properties of V1 L2/3 DS cells and the differences between control and *Frmd7tm* by first decomposing the response matrix into two dimensions using PCA (Fig. 4a, b). Similar to in areas RL and PM, V1 L2/3 neurons sharing the same TF preference were locally clustered (Fig. 4c). We found grids where the fraction of neurons was decreased, increased or unchanged in *Frmd7tm* mice (Fig. 4e). Neurons that were decreased in *Frmd7tm* mice increased their response amplitude as the TF increased ($P < 0.001$ for 0.3 versus 1.2 Hz for control and *Frmd7tm* mice, Wilcoxon signed-rank; Fig. 4f). In contrast, neurons that were increased in *Frmd7tm* mice showed a TF-dependent decrease in responses ($P < 0.001$ for 0.3 versus 1.2 Hz for control and *Frmd7tm* mice, Wilcoxon signed-rank), while neurons that were unchanged showed no TF-dependent response modulation ($P \geq 0.05$ for 0.3 versus 1.2 Hz for control and *Frmd7tm* mice, Wilcoxon signed-rank; Fig. 4f). Analyzing preferred motion directions for individual V1 L2/3 neurons at 0.3 and 1.2 Hz revealed that neurons that were decreased in *Frmd7tm* mice exhibited a direction tuning shift: at 0.3 Hz neurons uniformly encoded all directions, while at 1.2 Hz they notably preferred posterior motion and this was paralleled with an increase in DSI ($P < 0.001$ for 0.3 versus 1.2 Hz for control and *Frmd7tm* mice, Wilcoxon signed-rank; Fig. 4g). In contrast, neurons that were increased or unchanged in *Frmd7tm* mice showed no obvious direction tuning bias at either low or high TFs, but neurons that were increased in *Frmd7tm* mice decreased their DSI ($P < 0.001$ for 0.3 versus 1.2 Hz for control and *Frmd7tm* mice, Wilcoxon signed-

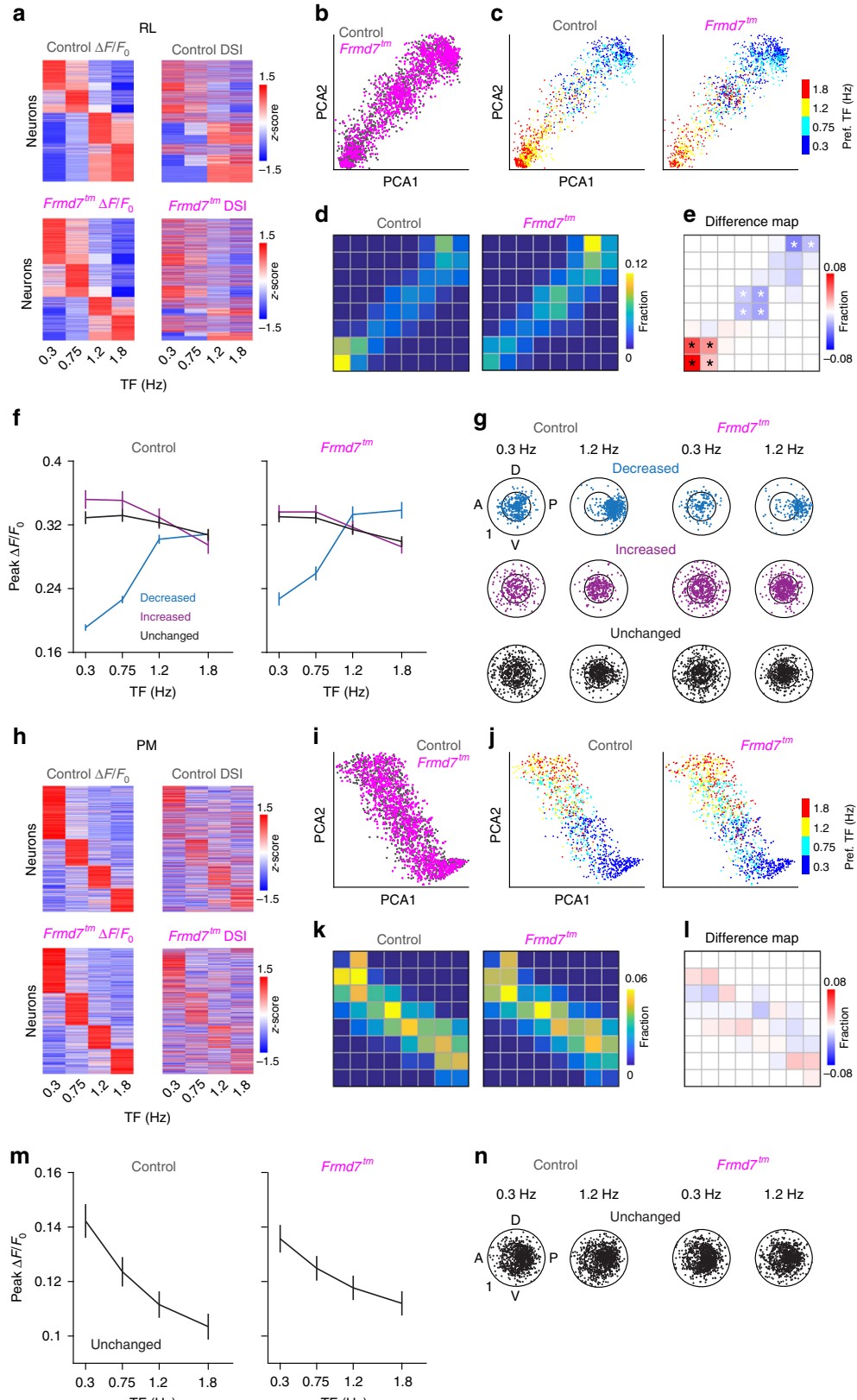

rank; Fig. 4g). Finally, we investigated the relationship between the effects of altered retinal horizontal direction selectivity and the target region of V1-projecting DS cells. We analyzed the fraction of PM- and RL-projecting neurons in each of the significantly affected grids. Importantly, this showed that the grids enriched in RL- and PM-projecting DS cells tend to be decreased and increased in

$Frmd7^{tm}$ mice, respectively (Fig. 4h). These results suggest that the impact of disrupted retinal horizontal direction selectivity in L2/3 of V1 is biased to a subset of neurons that prefer high-TF motion and preferentially encode posterior motion at high TFs. Furthermore, this subset of V1 neurons appeared to preferentially project to area RL.

**Fig. 2 Neurons with distinct functional characteristics are sensitive to disruption of retinal horizontal direction selectivity in area RL. a** Response matrix composed of TF-dependent response amplitudes and DSI for RL L2/3 DS cells sorted by TF preference. **b** Two-dimensional (2D) visualization of the 1st and 2nd principal components for the response matrix shown in **a**. Each point represents one neuron. **c** TF preference of individual RL neurons. **d** Fraction of neurons in 8 × 8 grids (gray lines) calculated from the PCA plot shown in **b**. **e** Fraction difference map between control and *Frmd7tm* mice. Black and white asterisks: significantly decreased and increased fractions in *Frmd7tm* mice, respectively, $P < 0.05$, two-sided $\chi^2$ test with Yates correction. **f** Peak response amplitude as a function of TF for three groups (decreased, increased, or unchanged in *Frmd7tm* mice) in control (572, 304, and 575 DS cells, respectively) and *Frmd7tm* mice (219, 508, and 659 DS cells, respectively). Error bars are SEM. **g** TF-dependent tuning characteristics of individual RL neurons from the three groups in control and *Frmd7tm* mice. Angular coordinate: preferred direction. Radial coordinate: DSI. Inner circle: DSI of 0.5. **h** Response matrix composed of TF-dependent response amplitudes and DSI for PM L2/3 DS cells sorted by TF preference. **i** 2D visualization of the 1st and 2nd principal components for the response matrix shown in **h**. Each point represents one neuron. **j** TF preference of individual PM neurons. **k** Fraction of neurons in 8 × 8 grids (gray lines) calculated from the PCA plot in **i** for control and *Frmd7tm* mice. **l** Fraction difference map between control and *Frmd7tm* mice. **m** Peak response amplitude as a function of TF for the unchanged group in control (1098 DS cells) and *Frmd7tm* mice (1217 DS cells). Error bars are SEM. **n** TF-dependent tuning characteristics of individual PM neurons from the unchanged group in control and *Frmd7tm* mice. Angular coordinate: preferred direction. Radial coordinate: DSI. Inner circle: DSI of 0.5. Source data are provided as a Source Data file.

**V1 L4 DS cells are insensitive to retinal manipulation.** Signaling from the retino-geniculate pathway is conveyed to L2/3 of V1 mainly via L4 (ref. [32]), and feature-tuned V1 L2/3 neurons receive L4 inputs[33]. Here, we investigated whether retina-originating DS signaling is routed to V1 L2/3 via L4, in addition to a previously suggested dLGN to V1 L1/2 shortcut pathway[23]. We imaged GCaMP6f-labeled neurons in L4 of V1 in anaesthetized mice (Supplementary Fig. 8), and identified DS cells from control and *Frmd7tm* mice (Fig. 5a). In contrast to L2/3, L4 DS cells from control and *Frmd7tm* mice were quantitatively very similar, both preferring 0.75 Hz TF, and showing no significant response amplitude differences (Fig. 5b, c). In control mice, the preferred directions of L4 DS cells showed a posterior bias at 0.3 and 1.2 Hz ($P < 0.05$, Rayleigh test; Fig. 5d), and this effect was not significantly different in *Frmd7tm* mice (Fig. 5d, e). The insensitivity of L4 DS cells to disruption of retinal horizontal direction selectivity at the population level was also supported by decomposition and segmentation analyses (Fig. 5f–h): none of the grids showed significantly different fractions between control and *Frmd7tm* mice (Fig. 5i, g). V1 L4 neurons showed no obvious change in direction tuning bias as a function of increasing TF, but neurons in both control and *Frmd7tm* mice increased their DSI ($P < 0.01$ for 0.3 versus 1.2 Hz for control and *Frmd7tm* mice, Wilcoxon signed-rank; Fig. 5i). These data show that DS cells in L4 of V1 are not noticeably sensitive to disruption of retinal horizontal direction selectivity, suggesting that retina-originating DS signaling reaches L2/3 of V1 via a pathway bypassing L4.

**DS thalamic axons are affected by retinal manipulation.** We examined whether the coupling of a high-TF preference and the TF-dependent development of posterior motion preference, which relies on retinal horizontal direction selectivity in areas V1 and RL, is already established in dLGN neurons. For this, we transfected dLGN neurons with GCaMP6f and imaged axons in L1 and L2/3 of V1 of anaesthetized mice (Fig. 6a). During grating stimuli, fluorescence increased in individual micron-sized varicosities along the axonal arborizations (Fig. 6b, c), confirming they were putative presynaptic boutons[13,23]. The fraction of DS boutons preferring high and low TFs decreased and increased, respectively, in *Frmd7tm* mice (Fig. 6e). In control mice, DS boutons preferred higher TFs in response to horizontal motion (Fig. 6f). In contrast, in *Frmd7tm* mice, DS boutons preferred lower TFs in horizontal directions, showing higher responses than control mice at 0.3 Hz and markedly lower responses at 1.2 and 1.8 Hz (Fig. 6f). Responses to vertical motion were unaltered in control and *Frmd7tm* mice (Fig. 6f). In control mice, the fraction of horizontally tuned DS boutons was significantly higher than vertically tuned boutons at both 0.3 and 1.2 Hz (Fig. 6g), whereas the fraction of vertically tuned DS boutons in *Frmd7tm*

mice was significantly higher at 1.2 Hz than horizontally tuned boutons (Fig. 6g). Thus, in control mice, the preferred directions of DS boutons showed a TF-invariant bias for posterior motion ($P < 0.001$ for 0.3 and 1.2 Hz, Rayleigh test; Fig. 6h), whereas they were biased to ventral at 1.2 Hz in *Frmd7tm* mice ($P \geq 0.05$ and $P < 0.01$ for 0.3 and 1.2 Hz, respectively; Rayleigh test; Fig. 6h, i). In control mice, 36.9% of DS boutons increased response amplitudes as the TF increased from 0.3 to 1.2 Hz, whereas only 17.1% increased in *Frmd7tm* mice (Fig. 6j). Noticeably, posteriorly tuned DS boutons in control mice showed larger TF-dependent response increments than boutons preferring other directions (Fig. 6j, k). Conversely, DS boutons from *Frmd7tm* mice decreased their responses as the TF increased in all preferred directions (Fig. 6k), although posteriorly tuned boutons did show a smaller response decrement than anteriorly tuned boutons (Fig. 6k). These data suggest that two response characteristics of dLGN DS axons arriving in L1 and L2/3 of V1 were impaired in *Frmd7tm* mice: the preference for high-TF horizontal motion and the TF-invariant population bias for preferring posterior motion.

**Altered TF-dependent responses in the retina of *Frmd7tm* mice.** TF-dependent responses of retinal neurons in *Frmd7tm* mice have not previously been examined[25,27]. We sought to link the TF-dependent response modulation observed in thalamic axons and cortical neurons to that in retinal neurons. We performed two-photon calcium imaging from the ganglion-cell layer in isolated retinas by the virally transduced GCaMP6s (Fig. 7a–c). In control retinas, neurons preferred high-TF stimuli moving in posterior, anterior, and ventral directions (i.e., nasal, temporal, and superior on the retina, respectively), whereas we found no TF-dependent response modulation for dorsal motion (i.e., inferior on the retina; Fig. 7d). In *Frmd7tm* retinas, the high-TF preference of neurons was impaired for horizontal motion: their responses to higher TFs were significantly lower than control mice (Fig. 7d). In contrast, responses to vertical motion were unaltered, as has been shown previously[27].

Next, we restricted our analyses to retinal neurons showing both ON-OFF responses and DS tuning (36.7% and 35.8% of all responsive neurons in control and *Frmd7tm* retinas, respectively; Supplementary Fig. 9). As previously reported[27], *Frmd7tm* retinas showed significantly decreased ON-OFF DS responses to horizontal motion (Fig. 7e). In control mice, there was a population bias for posterior motion at 1.2 Hz ($P < 0.05$, Rayleigh test; Fig. 7f), whereas DS cells from *Frmd7tm* mice were biased towards vertical motion at both 0.3 and 1.2 Hz ($P < 0.001$ for 0.3 and 1.2 Hz, Rayleigh test; Fig. 7f, g). Interestingly, only posteriorly tuned ON-OFF DS cells showed larger TF-dependent response increments and a larger TF-dependent increase in DSI in control retinas ($P < 0.001$ and $P \geq 0.05$ for all comparisons in control and *Frmd7tm*

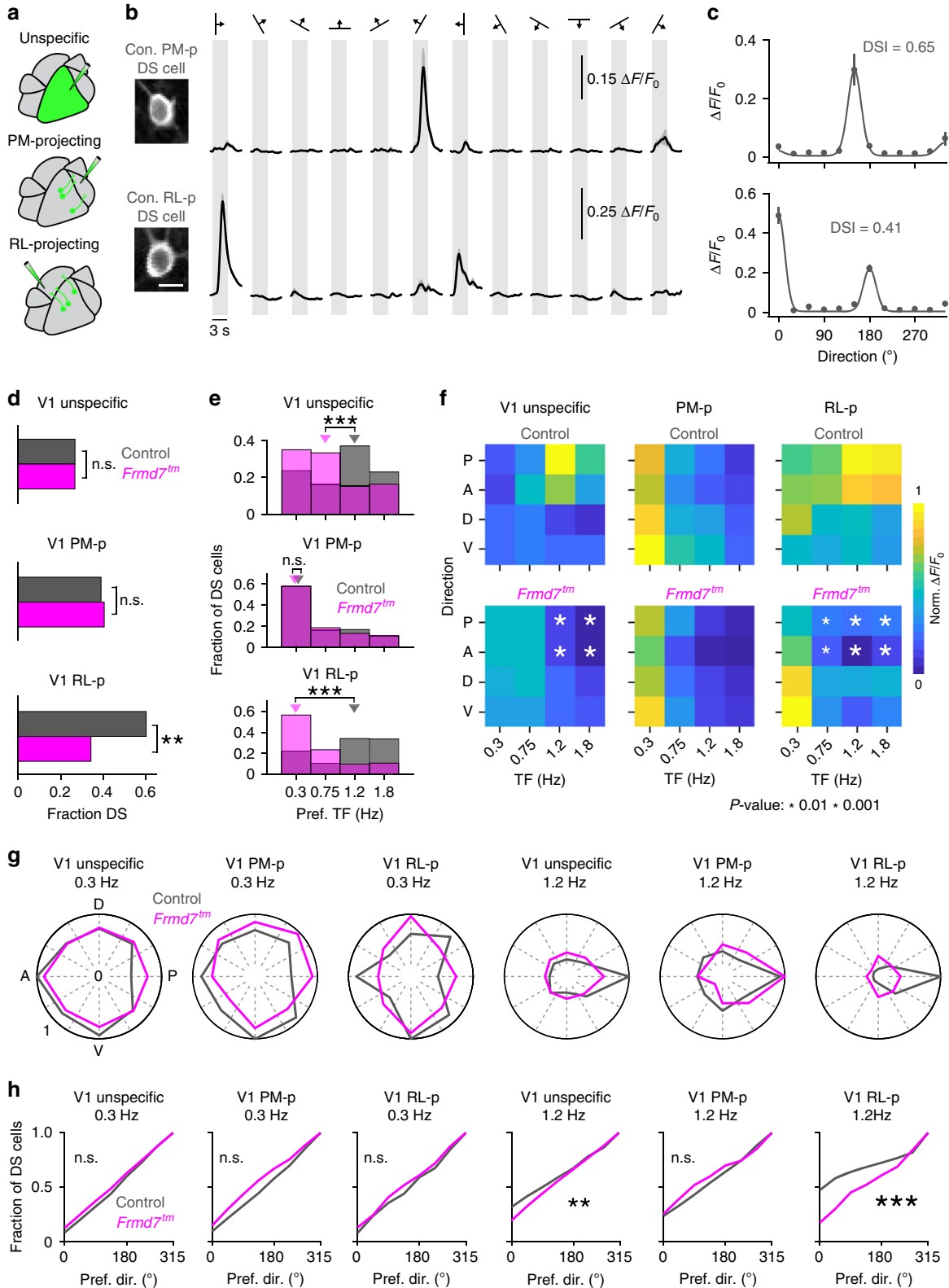

mice, respectively, Mann–Whitney $U$-test; Fig. 7h, i). These data suggest at least two mechanisms underlying the retinal over-representation of a preference for high-TF posterior motion: First, ON-OFF DS cells that prefer posterior motion are more abundant than other populations, which becomes more evident as the DSI of these cells increases at high TFs. Second, posteriorly tuned retinal DS cells show larger response increments than DS cells preferring

other directions as TF increases. Importantly, both of these posterior biases are markedly impaired in $Frmd7^{tm}$ retinas.

## Discussion

Our results provide three major insights into the functional organization of murine visual pathways for processing visual

**Fig. 3 V1 DS cells projecting to area RL or PM respond differently to disruption of retinal direction selectivity. a** Target-unspecific V1 neurons were labeled by injecting AAV2/1-GCaMP6f into the V1. PM- (PM-p) and RL-projecting (RL-p) V1 neurons were labeled by injecting rAAV2-retro-GCaMP6m into PM and RL, respectively. **b** Two-photon calcium imaging from L2/3 in V1 of control (1087 target-unspecific, 109 PM-p, and 513 RL-p DS cells; 5 mice per group) and *Frmd7tm* mice (954 target-unspecific, 93 PM-p, and 235 RL-p DS cells; five mice per group). Left: example PM- and RL-projecting neurons expressing GCaMP6m (scale bar, 10 μm). Right: trial-averaged fluorescence ($\Delta F/F_0$) time courses for the same neurons. Shading: SEM. **c** Tuning curves for neurons shown in **b**. Error bars: SEM. Solid line: Gaussian fit. **d** Fraction of DS cells in target-unspecific, PM-p, and RL-p V1 neuronal populations (two-sided $\chi^2$ test with Yates correction). **e** Preferred TF for target-unspecific (two-sided Mann–Whitney *U*-test), PM-p (two-sided Mann–Whitney *U*-test), and RL-p (two-sided Mann–Whitney *U*-test) V1 DS cells. Triangles: Medians. **f** Response amplitude as a function of motion direction and TF for target-unspecific, PM-p, and RL-p V1 DS cells. White asterisks: significantly decreased in *Frmd7tm* mice, two-sided Mann–Whitney *U*-test. **g** Fractional distributions of preferred motion directions for target-unspecific, PM-p, and RL-p V1 DS cells at 0.3 and 1.2 Hz. The fractions are normalized to the largest fraction across genetic groups. **h** Distributions of preferred motion directions at 0.3 and 1.2 Hz for target-unspecific, PM-p, and RL-p V1 DS cells (two-sided Kolmogorov–Smirnov test). **P < 0.01, ***P < 0.001, n.s., not significant, in **d**, **e**, and **h**. Source data are provided as a Source Data file.

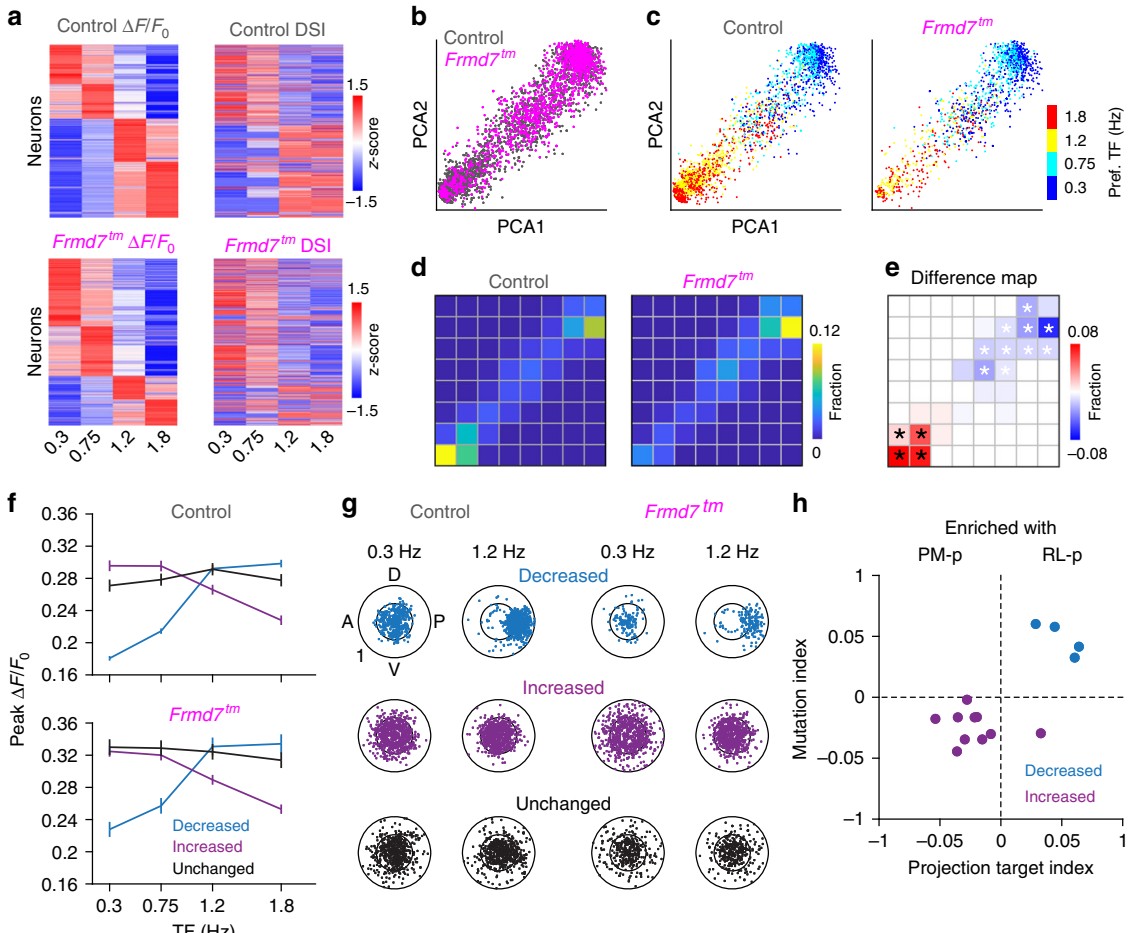

**Fig. 4 V1 L2/3 neurons with distinct functional characteristics are sensitive to disruption of retinal horizontal direction selectivity. a** Response matrix composed of TF-dependent response amplitudes and DSI for all pooled V1 L2/3 DS cells (target-unspecific, PM-p, and RL-p) sorted by TF preference. **b** Two-dimensional (2D) visualization of the 1st and 2nd principal components for the response matrix shown in **a**. Each point represents one neuron. **c** TF preference of individual V1 neurons. **d** Fraction of neurons in 8 × 8 grids (gray lines) calculated from the PCA plot shown in **b** for control and *Frmd7tm* mice. **e** Fraction difference map between control and *Frmd7tm* mice. Black and white asterisks: significantly decreased and increased fractions in *Frmd7tm* mice, respectively, *P* < 0.05, two-sided $\chi^2$ test with Yates correction. **f** Peak response amplitude as a function of TF for three groups (decreased, increased, or unchanged in *Frmd7tm* mice) in control (886, 845, and 591 DS cells, respectively) and *Frmd7tm* mice (169, 825, and 381 DS cells, respectively). Error bars are SEM. **g** TF-dependent tuning characteristics of individual V1 neurons from the three groups in control and *Frmd7tm* mice. Angular coordinate: preferred direction. Radial coordinate: DSI. Inner circle: DSI of 0.5. **h** Relationship between effect of *Frmd7tm* mutation and enrichment in PM-p or RL-p neurons. *x*-axis: Index comparing axonal projection pattern for the individual grids in **e** that were decreased and increased in *Frmd7tm* mice; groups with positive and negative index values are enriched in RL- and PM-projecting neurons, respectively. *y*-axis: Index comparing sensitivity to altered retinal direction selectivity for grids; grids with a positive and negative index value are decreased and increased in *Frmd7tm* mice, respectively. Source data are provided as a Source Data file.

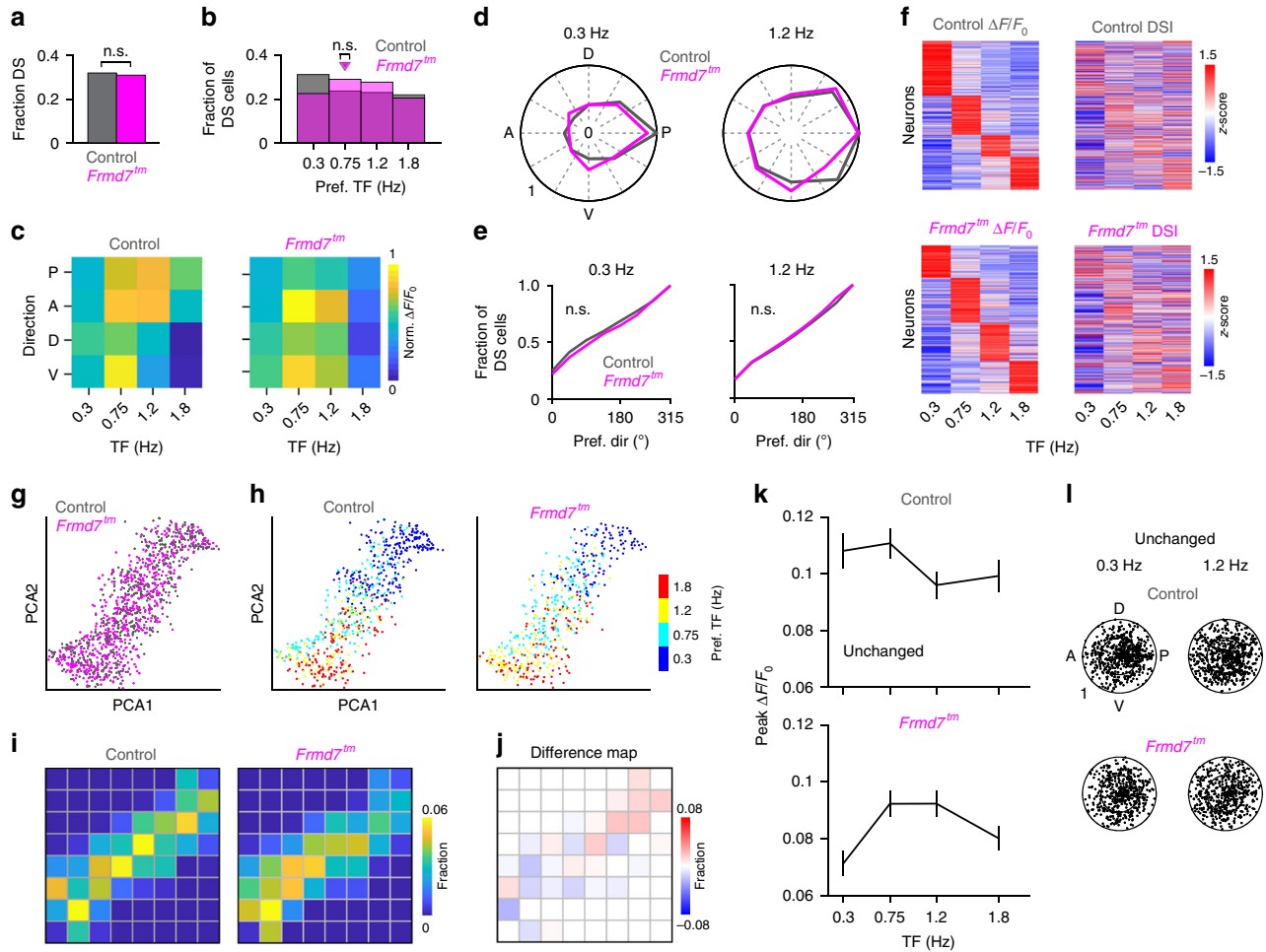

**Fig. 5 Direction selectivity in V1 L4 is insensitive to disruption of retinal horizontal direction selectivity. a** Fraction of DS cells in V1 L4 (678 and 536 DS cells in control and *Frmd7tm* mice, respectively; four mice per group; two-sided $\chi^2$ test with Yates correction). **b** Preferred TF for DS cells in V1 L4 (two-sided Mann–Whitney *U*-test). Triangles: Medians. **c** Response amplitude as function of motion direction and TF for V1 L4 DS cells. **d** Fractional distributions of preferred motion directions for V1 L4 DS cells at 0.3 and 1.2 Hz, normalized to the largest fraction across genetic groups. **e** Distributions of preferred motion directions at 0.3 and 1.2 Hz for V1 L4 DS cells (two-sided Kolmogorov–Smirnov test). **f** Response matrix composed of TF-dependent response amplitudes and DSI for V1 L4 DS cells sorted by TF preference. **g** Two-dimensional (2D) visualization of the 1st and 2nd principal components for the response matrix shown in **f**. Each point represents one neuron. **h** TF preference of individual V1 L4 neurons. **i** Fraction of neurons in 8 × 8 grids (gray lines) calculated from the PCA plot shown in **g** for control and *Frmd7tm* mice. **j** Fraction difference map between control and *Frmd7tm* mice. **k** Peak response amplitude as a function of TF for the unchanged group in control (678 DS cells) and *Frmd7tm* mice (536 DS cells). Error bars are SEM. **l** TF-dependent tuning characteristics of individual V1 L4 neurons from the unchanged group in control and *Frmd7tm* mice. Angular coordinate: preferred direction. Radial coordinate: DSI. Inner circle: DSI of 0.5. n.s., not significant. Source data are provided as a Source Data file.

motion. First, we identified area RL as a higher visual cortical area, where the response properties of DS cells prominently rely on retinal motion computations. Second, we identified a connection motif in the cortico-cortical V1 L2/3 projection neurons, by which feedforward signaling, originating from retinal DS cells, is selectively routed to area RL, but not to area PM (Fig. 7j). Third, retinal DS cells influence cortical neurons by biasing their direction tuning toward posterior motion in a stimulus TF-dependent manner. Our results thus point to an unexpected causal link between a specialized response feature of HVAs and a particular form of retinal computation.

Our work provides key insights into the neural circuit mechanisms enabling functional diversity in HVAs. Examining the input-output relationship of the mouse V1 has been challenging, since distinct projection neurons do not have any known histological characteristics, unlike their counterparts in primates where V1 neurons mediating the magnocellular and parvocellular pathways can be distinguished by histological features[34,35].

Recent research has demonstrated that the connections between projection neurons in V1 that project to different targets are rare, regardless of response similarities[16]. Thus, our results suggest that each of these segregated subnetworks potentially receives a unique combination of retino-geniculate inputs, enabling specialized responses of individual HVAs to emerge. Our findings invoke the intriguing question of how such specific multi-synaptic connectivity may be established during development. One possibility is that downstream circuits of retinal DS cells are wired together based on a set of uniquely expressed molecules. Alternatively, or synergistically, mechanisms dependent on patterned spontaneous activity[36] or visual experience may guide the synaptic connections.

By analyzing the PCA distributions of the imaged neurons, we identified that the coupling of response amplitude increments and the development of a posterior motion preference as the stimulus TF increases are the key response features in RL and V1 L2/3 DS cells that rely on retinal motion computations along the

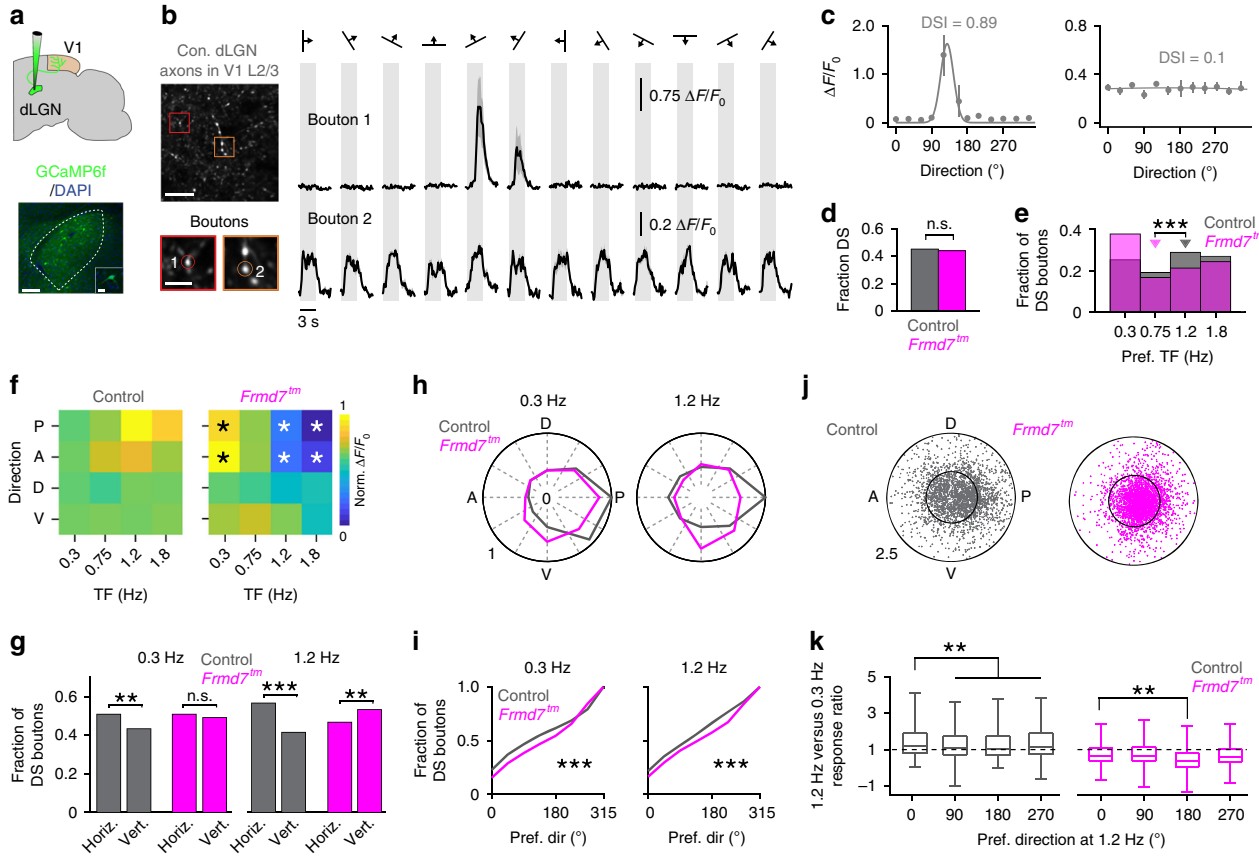

**Fig. 6 Directionally tuned thalamic boutons in the superficial V1 are sensitive to disruption of retinal horizontal direction selectivity. a** Top: thalamocortical axons in V1 were labeled by injecting AAV2/1-GCaMP6f into the dLGN. Bottom: Example GCaMP6f-positive neurons in dLGN (scale bar, 100 μm; inset, 10 μm). **b** Left top: Example two-photon image of dLGN axons in L2/3. Left bottom: magnified axonal boutons expressing GCaMP6f (scale bar, 35 μm; inset, 10 μm). Right: trial-averaged fluorescence ($\Delta F/F_0$) time courses for the boutons indicated. Shading: SEM. **c** Tuning curves for boutons shown in **b**. Error bars: SEM. Solid line: Gaussian fit. **d** Fraction of DS boutons (5,121 and 5,525 DS boutons from control and *Frmd7tm* mice, respectively; five mice per group; two-sided $\chi^2$ test with Yates correction). **e** Preferred TF for DS boutons (two-sided Mann–Whitney U-test). Triangles: Medians. **f** Response amplitude as a function of motion direction and TF for DS boutons. White and black asterisks: significantly decreased and increased responses, respectively, in *Frmd7tm* mice, $P < 0.05$, two-sided Mann–Whitney U-test. **g** Fraction of horizontally- and vertically tuned DS boutons at 0.3 and 1.2 Hz (two-sided $\chi^2$ test with Yates correction). **h** Fractional distributions of preferred motion directions for DS boutons at 0.3 and 1.2 Hz, normalized to the largest fraction across genetic groups. **i** Distribution of preferred motion directions at 0.3 and 1.2 Hz in DS boutons (two-sided Kolmogorov–Smirnov test). **j** TF-dependent characteristics of DS boutons. Angular coordinate: Directional preference at 1.2 Hz. Radial coordinate: Ratio of response amplitudes at 1.2 and 0.3 Hz. Inner circle: Response ratio of 1. **k** Ratio of response amplitudes at 1.2 and 0.3 Hz as a function of motion direction preference of DS boutons (two-sided Mann–Whitney U-test). Center line is median, box limits are 25th and 75th percentiles, and whiskers show minimum and maximum values. *$P < 0.05$, **$P < 0.01$, ***$P < 0.001$, n.s., not significant. Source data are provided as a Source Data file.

horizontal axis. How do such TF-dependent changes in responses and tuning rely on retinal motion computations? We propose that the retinal over-representation of posterior motion that develops with increasing TF, and which is conveyed via the dLGN, is amplified by the local circuitry within V1, shifting the balance of cortical DS responses to the posterior direction. The suppression of responses to non-posterior directions at higher TFs could result from cortical normalization[37]. The larger retinal population responses to posterior compared to anterior motion at higher TFs may be advantageous for distinguishing rotational and translational optic flow by enabling differences of summed output activity from left versus right retinas. The higher TF sensitivity of horizontal compared to vertical retinal DS cells may be explained by the two-dimensional nature of terrestrial navigation in mice. In contrast to RL and V1 L2/3 neurons, direction selectivity in V1 L4 and PM L2/3 neurons is most likely generated by non-retinal mechanisms[26] (Figs. 1 and 5), indicating spatially segregated processing of direction selectivity computed by retina-dependent and -independent mechanisms. Lastly, it is worth noting that we

cannot exclude the possibility that signaling from retinal DS cells is also conveyed to area RL via extrageniculate pathways[38–40], given the innervation of the superior colliculus by retinal ON-OFF DS cells[41], despite collicular outputs preferentially target the postrhinal cortex[15].

What is the possible behavioral role of area RL and retinal ON-OFF DS cells? In mice, area RL is thought to belong to a dorsal-like stream[7,11,12,38], and the anterior part of RL is considered to be part of the posterior parietal cortex[42,43], which is important for visually guided navigation[44,45]. Furthermore, area RL is strongly interconnected with other areas such as area AL, the primary somatosensory cortex, and the secondary motor cortex[46]. Notably, >50% of RL L2/3 pyramidal neurons are multisensory, integrating both tactile and visual inputs[47], and RL is also thought to be involved in visuo-motor integration[48]. Lastly, the receptive field location of RL neurons is biased to the anterior, lower visual-field[8] and the neurons respond to visual stimuli very close to the mouse[49]. Altogether, these prior findings indicate that area RL is ideally adapted to sensori-motor coordination for the lower

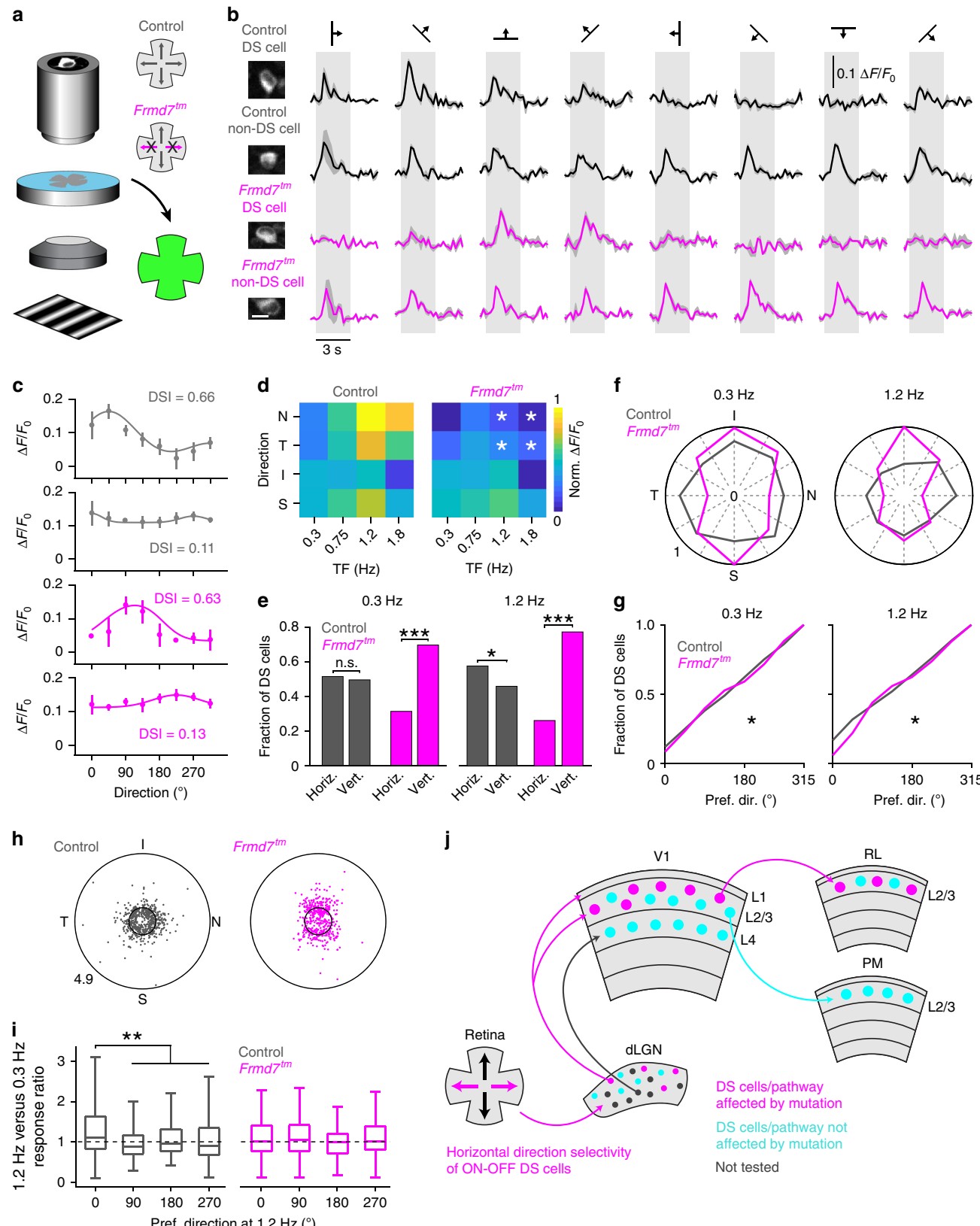

visuo-tactile space near the face of the mouse. Interestingly, the ventral intraparietal area of the human posterior parietal cortex, which is referred to as the dorsal stream of vision, also contains visual and tactile maps, and is focused on processing the face-centered sensory space[50]. In macaque monkeys, this area contains many DS cells, which prefer high speeds, and some of these cells are sensitive to the trajectory of visual stimuli moving toward the face[51]. Furthermore, this area jointly represents translation direction and rotation velocity during self-motion based on optic flow[52]. Together with a recent finding that preferred directions of

**Fig. 7 Preference of retinal neurons to posterior motion at higher TFs is disrupted in *Frmd7tm* mice. a** Two-photon calcium imaging was performed on retinas from control (1157 cells; four mice) and *Frmd7tm* mice (953 cells; four mice). **b** Left: example control and *Frmd7tm* retinal neurons expressing GCaMP6s (scale bar, 10 μm). Right: trial-averaged fluorescence ($\Delta F/F_0$) time courses for the same. Shading indicates SEM. **c** Tuning curves for neurons shown in **b**. Error bars are SEM. Solid line: Gaussian fit. **d** Response amplitude as a function of motion direction (nasal [N], temporal [T], inferior [I], and superior [S]) and TF for retinal cells. White asterisk: significantly decreased response in *Frmd7tm* mice (two-sided Mann–Whitney *U*-test). **e** Fraction of horizontally- and vertically-tuned retinal ON-OFF DS cells (425 and 342 ON-OFF DS cells in control and *Frmd7tm* mice, respectively) at 0.3 and 1.2 Hz (two-sided $\chi^2$ test with Yates correction). **f** Fractional distributions of preferred motion directions for ON-OFF DS cells at 0.3 and 1.2 Hz, normalized to the largest fraction across genetic groups. **g** Distribution of preferred motion directions at 0.3 and 1.2 Hz in ON-OFF DS cells (two-sided Kolmogorov–Smirnov test). **h** TF-dependent characteristics of ON-OFF DS cells. Angular coordinate: Directional preference at 1.2 Hz. Radial coordinate: Ratio of response amplitudes at 1.2 and 0.3 Hz. Inner circle: response ratio of 1. **i** Ratio of response amplitudes at 1.2 and 0.3 Hz as a function of motion direction preference of cells that showed ON-OFF DS responses (two-sided Mann–Whitney *U*-test). Center line is median, box limits are 25th and 75th percentiles, and whiskers show minimum and maximum values. **j** Schematic diagram of proposed neural pathway linking retinal ON-OFF DS cells to RL DS cells. *$P < 0.05$, **$P < 0.01$, ***$P < 0.001$, n.s., not significant. Source data are provided as a Source Data file.

retinal ON-OFF DS cells are aligned with translatory and rotatory optic flow fields[21], our findings raise the intriguing hypotheses that the mouse area RL may be a functional counterpart of the primate ventral intraparietal area, and visual motion analyses in the primate posterior parietal cortex may rely on signaling from such retinal DS cells. However, it is still unknown if dLGN-projecting retinal DS cells exist in primates.

## Methods

**Experimental animals**. Wild-type control mice (C57BL/6J) were obtained from Janvier Labs. *Frmd7tm* mice are homozygous female or hemizygous male *Frmd7tm1b(KOMP)Wtsi* mice, which were obtained as *Frmd7tm1a(KOMP)Wtsi* from the Knockout Mouse Project (KOMP) Repository[25,27]: Exon 4 and neo cassette flanked by loxP sequences were removed by crossing with female Cre-deleter *Edil3Tg(Sox2−cre)1Amc/J* mice (Jackson laboratory stock 4783) as confirmed by PCR of genome DNA, and maintained in C57BL/6J background. *ChAT-Cre* (strain: *Chattm2(cre)Lowl*/MwarJ, Jackson laboratory stock: 028861)[53] and *LSL-DTR* (strain: *Gt(ROSA)26Sortm1(HBEGF)Awai*/J, Jackson laboratory stock: 007900)[54] were purchased from Jackson laboratory and maintained in C57BL/6J background. Experiments were performed on 34 male and female wild-type control mice, 33 male and female *Frmd7tm* mice, and 10 *ChAT-Cre × LSL-DTR* mice. All mice were between two and four months old. Mice were group-housed and maintained in a 12-h/12-h light/dark cycle with ad libitum access to food and water. Experiments were performed according to standard ethical guidelines and were approved by the Danish National Animal Experiment Committee.

**Head-plate and cranial window implantation**. Mice were anaesthetized with an intraperitoneal injection of a fentanyl (0.05 mg/kg body weight; Hameln), midazolam (5.0 mg/kg body weight; Hameln) and medetomidine (0.5 mg/kg body weight; Domitor, Orion) mixture dissolved in saline. The depth of anesthesia was monitored by the pinch withdrawal reflex throughout the surgery. Core body temperature was monitored using a rectal probe and temperature maintained at 37−38 °C using a feedback-controlled heating pad (ATC2000, World Precision Instruments). Eyes were protected from dehydration during the surgery with eye ointment (Oculotect Augengel). The scalp overlaying the left visual cortex was removed, and a custom head-fixing imaging head-plate with a circular 8 mm diameter opening was mounted on the skull using cyanoacrylate-based glue (Super Glue Precision, Loctite) and dental cement (Jet Denture Repair Powder) to allow for subsequent head fixation during imaging. The center of the head-plate was positioned above V1, 2.5 mm lateral and 1 mm anterior of lambda[55]. To gain optical access to the cortex, a 5 mm diameter craniotomy was performed. After removing the skull flap, the cortical surface was kept moist with a cortex buffer containing 125 mM NaCl, 5 mM KCl, 10 mM glucose, 10 mM HEPES, 2 mM MgSO$_4$, and 2 mM CaCl$_2$. The dura was left intact (except in two animals in which the dura spontaneously detached with the skull flap) and any occasional bleedings were immediately stopped with Gelfoam (Pfizer). A 5 mm glass coverslip sterilized in ethanol (0.15 mm thickness, Warner Instruments) was placed onto the brain to gently compress the underlying cortex and dampen biological motion during subsequent imaging[56]. The cranial window was hermetically sealed using a cyanoacrylate-based glue (Super Glue Precision, Loctite) mixed with black dental cement (Jet Denture Repair Powder mixed with iron oxide powdered pigment) to prevent the entry of stray light from the screen through the skull and/or cement during imaging[56]. Mice were returned to their home cage after anesthesia was reversed with an intraperitoneal injection of a flumazenil (0.5 mg/kg body weight; Hameln) and atipamezole (2.5 mg/kg body weight; Antisedan, Orion Pharma) mixture dissolved in saline, and after recovering on a heating pad for one hour.

**Intrinsic imaging**. For ISOI mice were anesthetized with isoflurane (2−3% induction) and head-fixed in a custom holder. Chlorprothixene was administered intraperitoneally (2.5 mg/kg body weight; Sigma) as a sedative[12], and isoflurane reduced to 0.5−1% and kept constant during visual stimulation. Core body temperature was maintained at 37−38 °C using a feedback-controlled heating pad (ATC2000, World Precision Instruments). The stimulated contralateral eye was kept lubricated by hourly application of a thin layer of silicone oil (OFNA Racing, 10,000 molecular weight). The experimental setup employed for ISOI was adapted from a similar system[57] and made publicly available (https://snlc.github.io/ISI/). A 2 × air-objective (Olympus, 0.08 NA, 4 mm field of view) was mounted on our Scientifica VivoScope, which was equipped with a CMOS camera (HD1-D-D1312-160-CL-12, PhotonFocus), a large-well-depth camera that offers high signal-to-noise measurements in bright light conditions. The camera was connected to a Matrox Solios (eCL/XCL-B) frame-grabber via Camera Link. The acquisition code for the Matrox board was written in Matlab using the Image Acquisition Toolbox. From the pial surface, the microscope was defocused down 400−600 μm, where intrinsic signals were excited using a red LED (KL1600, Schott) delivered via light guides through a 610 nm long-pass filter (Chroma). Reflected light was captured through a 700 ± 50 nm (mean ± SEM) band-pass filter (Chroma) positioned right in front of the camera at a rate of 6 frames per second (512 × 512 pixels). At 700 nm there is a large change in the absorption coefficient between oxyhemoglobin and deoxyhemoglobin, contributing to the intrinsic signal measured in these experiments[12]. The 47.65 × 26.87 cm (width × height) screen was angled 30° from the mouse's midline and positioned so that the perpendicular bisector was 10 cm from the bottom of the screen, centered on the screen left to right (23.8 cm on each side), and 10 cm from the eye[57]. This resulted in a visual-field coverage from −41.98° to 60.77° (total 102.75°) in elevation and from −67.23° to 67.23° (total 134.46°) in azimuth. Thus, the stimulus covered almost the entire known visual hemi-field of the mouse, which is estimated to be at most 110° vertically and 140° horizontally. For the ISOI experiments presented in Fig. 1, each mouse was imaged on three separate days, and the data averaged to reduce the chance of day-to-day variations confounding group-level results.

**Visual stimuli for ISOI**. Retinotopic maps were generated by sweeping a spherically corrected (Matlab code provided by Spencer Smith: https://labrigger.com/blog/2012/03/06/mouse-visual-stim/) full-field bar across the screen in both azimuth and elevation directions[57]. The bar contained a flickering black-and-white checkerboard pattern on a black background[7,57]. The width of the bar was 12.5° and the checkerboard square size was 25°. Each square alternated between black and white at 4 Hz. In each trial, the bar was drifted 10 times in each of the four cardinal directions (0°, 90°, 180°, and 270°), moving at 8−9°/s. Usually, two to three trials were sufficient to achieve well-defined retinotopic maps. For measuring and characterizing the evoked areal activity in V1 and HVAs we presented 100% contrast black and white sinusoidal gratings drifting in each of the four cardinal directions. We presented gratings with four different TFs: 0.3, 0.75, 1.2, and 1.8 Hz (0.03 cycles/°). Each stimulus was stationary for 10 s and in motion for 10 s, comprising a stimulus period of 20 s, which was repeated 5 times in each direction. All stimuli used for ISOI were produced and presented using Matlab and the Psychophysics Toolbox[58].

**Image analysis for ISOI**. To generate functional visual cortex maps from the raw image data, we took the response time course for each pixel and computed the phase and magnitude of the Fourier transform at the stimulus frequencies (0.067 and 0.088 Hz, for azimuth and vertical, respectively)[59]. The bar was drifted in opposite directions in order to subtract the delay in the intrinsic signal relative to neuronal activity[59]. The resulting phase maps were then converted into retinotopic coordinates (visual degrees) from the known geometry of our setup to retrieve absolute retinotopy. We used automated, publicly available code to identify visual area borders based on their visual-field sign maps[57] (see also description below),

and superimposed those borders with the anatomical blood-vessel images to accurately localize V1 and individual HVAs.

To evaluate and quantify the spatial properties of visual areas, we first computed and identified the borders of each visual area, using a dissociation algorithm and the Image Processing Toolbox in Matlab. The identification process constituted three conventional edge-detection steps. (1) Thresholding of the obtained visual-field sign map. For this we used a definition of $\bar{I} + SD(I)$ where $I$ equals the intensity of a pixel and $\bar{I}$ denotes the mean pixel intensity of the visual-field sign map. The thresholded image was smoothed using a median filter (filter size, $3 \times 3$ neighborhoods). (2) Isolation and segmentation of pixels. For this step we used the 8-neighbors criterion. If there were >4 non-zero pixels among the eight neighbors of one pixel, the pixel was retained and any gaps between pixels within the eight neighbors were filled with a non-zero value. If this criterion was not met, the pixel value was set to 0. The value of all non-zero pixels was set to 1. (3) Edge detection. The isolated and segmented pixels were next binarized, and each edge was computed based on a Sobel method using the edge function in Matlab. After identifying visual area borders, we computed the size of the area (first in pixels and then converted into mm2) and defined the center position of each area as a centroid. From this, we calculated two-dimensional coordinates of each HVA as coordinates relative to the V1 centroid. For ISOI experiments in which sinusoidal gratings were presented, the raw response signal was first determined as the peak power of the stimulus-evoked signal by employing Fast-Fourier transform analyses of each pixel column at the frequencies of the visual stimuli; $0.05-0.1 Hz$[12]. To quantify the response for each visual cortical area, the raw response signal was first normalized to the raw response signal from before the visual stimulation (averaged over a 10 s period). Next, regions of interest (ROIs) within the visual cortical areas were defined based on the visual area border map. The response strength of each area was determined as the maximum value within each ROI. For group-level quantifications, the response strength for a given area was averaged across the three experimental days before pooling data from all mice.

**Local viral labeling**. For local viral injections in the visual cortex or dLGN, mice were first anesthetized with an intraperitoneal injection of a fentanyl (0.05 mg/kg body weight; Hameln), midazolam (5.0 mg/kg body weight; Hameln) and medetomidine (0.5 mg/kg body weight; Domitor, Orion) mixture dissolved in saline. For injections yielding GCaMP6 expression in areas V1, RL and PM, three 0.4 mm diameter craniotomies were performed over the left visual cortex and 100 nl AAV2/1-Syn-GCaMP6f-WPRE ($2.13 \times 10^{13}$ vg/ml, Penn Vector Core #AV-1-PV2822) slowly injected at depths of $100-500$ μm using a borosilicate glass micropipette (30 μm tip diameter) and a pressure injection system (Picospritzer III, Parker). For labeling geniculo-cortical axons projecting from the dLGN, $20-40$ nl AAV2/1-Syn-GCaMP6f-WPRE ($2.13 \times 10^{13}$ vg/ml, Penn Vector Core #AV-1-PV2822) was slowly injected into the left dLGN using stereotaxic coordinates: 2.1 mm posterior of the Bregma; 2.2 mm lateral of the midline; 2.3 mm below the pial surface[23]. To prevent backflow during withdrawal, the micropipette was kept in the brain for a minimum of 5 min before it was slowly retracted. The skin was afterwards sutured shut. Mice were returned to their home cage after the anesthesia was reversed with an intraperitoneal injection of a flumazenil (0.5 mg/kg body weight; Hameln) and atipamezole (2.5 mg/kg body weight; Antisedan, Orion Pharma) mixture dissolved in saline, and after recovering on a heating pad for 1 h. Although the injection sites in the visual thalamus were always within the dLGN, there were sometimes spillover expressions in the neighboring ventral LGN (vLGN), intergeniculate leaflet (IGL), and the pulvinar nucleus similar to previously reported[60]. The vLGN and IGL do not project to V1 (ref. [61]); compared to the dLGN, projections from the pulvinar to V1 are much sparser and limited to the superficial L1 (ref. [62]). Hence, the majority of thalamic axons that we imaged in V1 most likely originated from the dLGN.

**Retrograde viral labeling**. To achieve GCaMP6 expression in V1 neurons projecting to area RL or PM we employed a slightly modified surgery protocol. First, we implanted a custom head-fixing imaging head-plate and mapped the visual cortex using ISOI through the intact skull. This allowed us to identify the precise anatomical location of areas RL and PM. Next, we performed a single, local virus injection into either area RL or PM by slowly injecting 100 nl of ssAAV-retro/2-hSyn1-mRuby2-GCaMP6m-WPRE ($7 \times 10^{12}$ vg/ml, VVF Zurich #v187-retro) at depths of $100-300$ μm. This AAV-retro serotype permits selective retrograde labeling of projection neurons and enables sufficient expression for functional two-photon calcium imaging[63]. After slowly retracting the micropipette, the craniotomy was carefully cleaned and the exposed skull covered with a silicone sealant (Kwik-Cast, World Precision Instruments). One day later, the animal was implanted with a chronic cranial window, exposing V1 for two-photon calcium imaging targeted to V1 neurons projecting specifically to either the RL or PM area.

**Diphtheria toxin injections**. To abolish retinal direction selectivity acutely in adult mice, we injected diphtheria toxin intravitreously into *ChAT-Cre × LSL-DTR* mice[25]. Diphtheria toxin stock solution was made from diphtheria toxin (Sigma, D0564) dissolved in phosphate-buffered saline (PBS) to a concentration of 1 μg/μl and stored at $-80$ °C. Before injections, the stock solution was diluted in PBS to a final concentration of 0.8 ng/μl. Mice were first anesthetized with an intraperitoneal

injection of a fentanyl (0.05 mg/kg body weight; Hameln), midazolam (5.0 mg/kg body weight; Hameln) and medetomidine (0.5 mg/kg body weight; Domitor, Orion) mixture dissolved in saline. A hole was made near the border between the sclera and the cornea using a 30-gaugle needle; 2 μl toxin was then injected into the vitreous of both eyes using a borosilicate glass micropipette connected to a pressure injection system (Picospritzer III, Parker). Mice were returned to their home cage after the anesthesia was reversed with an intraperitoneal injection of a flumazenil (0.5 mg/kg body weight; Hameln) and atipamezole (2.5 mg/kg body weight; Antisedan, Orion Pharma) mixture dissolved in saline. Each eye was re-injected 2 days after the initial injection. OMR recordings were performed $7-9$ days after the initial injection, and in-vivo two-photon calcium imaging experiments were initiated $10-12$ days after the initial injection.

**Optomotor response measurement**. For recording the optomotor reflex the mouse was placed on a central, raised platform and presented visual stimuli in the form of drifting sinusoidal gratings projected onto a virtual cylinder on the four surrounding computer screens[64]. The gratings were drifting horizontally at 12°/s, alternating the drift direction every 60°. One trial consisted of six 1 min repeats; after each repeat, the spatial frequency of the stimulus was sequentially changed (0.05, 0.1, 0.2, 0.25, 0.3, 0.4 cycles/°). Mouse head movements were tracked using OKR arena software[64], where the angle of the head is automatically calculated and used to quantify the OMR for each stimulus condition[64]. OMR was determined by calculating the ratio of the sum of frames where head movements occurred in the stimulus direction versus in the opposite direction[64].

**Cortical two-photon calcium imaging**. Imaging was performed $2-4$-weeks after virus injections, when most neurons exhibited cytosolic-only GCaMP6 expression. Mice were anesthetized with 0.3–0.8% (typically 0.5%) isoflurane, and chlorprothexine was delivered intraperitoneally (2.5 mg/kg body weight; Sigma) as a sedative[12]. The stimulated contralateral eye was kept lubricated by hourly application of a thin layer of silicone oil (OFNA Racing, 10,000 molecular weight). Core body temperature was maintained at $37-38$ °C using a feedback-controlled heating pad (World Precision Instruments, ATC2000). A subset of experiments was performed in awake mice. To habituate the mice to handling and experimental conditions, each mouse was head-fixed onto the imaging stage with its body restrained in a cylindrical cover, reducing struggling and substantial body movements[60,65]. The habituation procedure was repeated for 3 days for each mouse at durations of 15, 30, and 60 min on day 1, day 2, and day 3, respectively. Mice were rewarded with chocolate paste (Nutella) at the end of each habituation/imaging session. For imaging, the mouse was placed under the microscope 10 cm from the 47.65 × 26.87 cm (width × height) screen, with the screen subtending 134.46° in azimuth and 102.75° in elevation and angled 30° from the mouse's midline. The visual area targeted for two-photon calcium imaging was localized based on superimposing the ISOI border map onto the cortical surface. Imaging was performed 50–100 μm (L1), 120–250 μm (L2/3), and 350–550 μm (L4) below the dura using a scanning microscope (VivoScope, Scientifica) with a 7.9 kHz resonant scanner running SciScan version 1.3 with dispersion-compensated 940 nm excitation provided by a mode-locked Ti:Sapphire laser (MaiTai DeepSee, Spectra-Physics) through either a Nikon 16× (0.8 NA; somata imaging) or an Olympus 25× (1.05 NA; axonal bouton imaging) objective. Clear ultrasound gel was used as an immersion medium (Aquasonic, Parker Laboratories). To prevent light leak originating from the visual stimulation, an imaging well was constructed from a black O-ring and the objective shielded with black tape. Average excitation power after the exit pupil of the objective varied from 25 to 60 mW. Typical images had 512 × 512 pixels, at $0.3-0.35$ μm per pixel for axons, and 0.92 μm per pixel for somata, and were acquired at 30.9 Hz using bidirectional scanning. By correcting for any slow drifts in neuron or axon location within the field of view using a reference image[6,23], we were able to record from the same population of neurons or axons over extended periods of time (~40 min), allowing us to assess responses as a function of TF conditions. There was no evidence of GCaMP6 bleaching during experiments. Each mouse was imaged repeatedly over the course of 1–2 weeks.

**Visual stimuli for cortical two-photon calcium imaging**. Visual stimulation for cortical two-photon calcium imaging was generated and presented via Python-based custom-made software. To measure directional tuning, we presented 100% contrast black and white sinusoidal drifting gratings. Drifting gratings were presented in six trials for 3 s at a time, with 3 s of gray screen between presentations, and were drifted in 12 different directions (0°, 30°, 60°, 90°, 120°, 150°, 180°, 210°, 240°, 270°, 300°, and 330°) in a pseudorandomized order, with a spatial frequency of 0.03 cycles/° and TFs of 0.3, 0.75, 1.2, and 1.8 Hz.

**Image analysis for cortical two-photon calcium imaging**. Imaging data were excluded from analysis if motion along the *z*-axis was detected. Raw images from somata imaging were corrected for in-plane motion via a correlation-based approach in Matlab[55]. Raw images from axonal bouton imaging were corrected for in-plane motion using a piecewise non-rigid motion correction algorithm[66]. ROIs were drawn in ImageJ (Cell Magic Wand; https://github.com/fitzlab/CellMagicWand) and selected based on mean and maximum fluorescence images[56]: somata ROIs were polygonal; axonal bouton ROIs were circular. The same

ROI set was used for all imaging stacks acquired in a given field of view. Fluorescence time courses were computed as the mean of all pixels within ROIs and were extracted using MIJ (http://bigwww.epfl.ch/sage/soft/mij/). Baseline-normalized fluorescence time courses ($\Delta F/F_0$) were computed using a 60 s 10th percentile filter and 0.01 Hz low-pass Butterworth filter to define $F_0$ (ref. [56]). For each of the twelve directions, the response amplitude in each trial was determined by sorting all $\Delta F/F_0$ values (down-sampled to 15.4 Hz) during the 3 s drift period, and taking the mean of the larger 50% of data points[25]. Somata and axonal boutons were defined as visually responsive if $\Delta F/F_0$ in the preferred direction of motion exceeded 0.06 (refs. [7,67]) in at least one of the four TFs. They were defined as DS if: (1) They were visually responsive; and (2) their DSI exceeded 0.3 (refs. [26,56]) in at least one of the four TFs:

$$\text{DSI} = \frac{R_{\text{pref}} - R_{\text{opp}}}{R_{\text{pref}} + R_{\text{opp}}}$$

where $R_{\text{pref}}$ denotes the mean $\Delta F/F_0$ response to the preferred direction of motion and $R_{\text{opp}}$ the mean $\Delta F/F_0$ response to the opposite direction. The preferred direction of motion for each cell was calculated as the angle, in polar coordinates, of the vector sum[27,68]:

$$\theta = \tan^{-1}\left(\frac{\sum_{i=1}^{12} R_i \sin d_i}{\sum_{i=1}^{12} R_i \cos d_i}\right)$$

where $d_i$ denotes the motion direction of direction $i$ and $R_i$ the mean $\Delta F/F_0$ response to direction $i$.

OSI was computed as:

$$\text{OSI} = \frac{R_{\text{pref}} - R_{\text{orth}}}{R_{\text{pref}} + R_{\text{orth}}}$$

where $R_{\text{pref}}$ denotes the mean $\Delta F/F_0$ response to the preferred orientation and $R_{\text{orth}}$ the mean $\Delta F/F_0$ response to the orthogonal orientation. The preferred orientation was defined as the axis including the preferred direction and its opposite direction.

**Data decomposition and segmentation**. To correlate the TF-dependent response properties in individual cortical DS cells and the fractional changes of neurons between control and *Frmd7tm* mice, we performed decomposition and segmentation of datasets summarizing response features of the identified DS cells. First, we composed a response matrix for each visual area (e.g., area RL) using a total of eight parameters for each neuron: peak $\Delta F/F_0$ amplitudes and DSI values under each of the four TF conditions. The response matrix included datasets from both control and *Frmd7tm* mice. For the V1 L2/3 response matrix, we pooled the datasets from target-unspecific, PM-projecting, and RL-projecting DS cells. Next, the response matrix was decomposed into two dimensions by PCA. The resulting PCA distributions showed a distribution trend depending on the TF preference of the individual neurons, indicating that neurons sharing the same TF preference tended to be clustered in the local region of the PCA distribution. To quantify the localization of neurons, we segmented the PCA distribution by 8 × 8 grids. We then calculated the fraction of neurons located within each of the grids, and examined the fractional changes between control and *Frmd7tm* mice. Based on the fractional changes, we statistically classified neurons into three groups: (1) "Increased", (2) "Decreased", and (3) "Unchanged" in *Frmd7tm* mice, compared to control mice. We tested the number of grids for this segmentation, and confirmed that the results were not qualitatively changed by the size of the grids (Supplementary Fig. 6).

To investigate the relationship between the effects of altered retinal horizontal direction selectivity and the target region of V1-projecting DS cells, we analyzed the fraction of PM- and RL-projecting cells in each grid based on a projection target index (PTI):

$$\text{PTI}^i = \frac{F_{\text{RLp}}^i - F_{\text{PMp}}^i}{F_{\text{RLp}}^i + F_{\text{PMp}}^i}$$

Where $F_y^x$ denotes the fraction of cells projecting to area $y$ in a grid $x$. A positive PTI indicates that the neurons within the grid are biased towards RL-projecting neurons, while a negative PTI indicates a bias towards PM-projecting neurons.

The effects of altered retinal horizontal direction selectivity was evaluated using a mutation index (MI):

$$\text{MI}^i = \frac{F_{\text{Control}}^i - F_{\text{Frmd7tm}}^i}{F_{\text{Control}}^i + F_{\text{Frmd7tm}}^i}$$

Where $F_y^x$ denotes the fraction of cells in population $y$ in grid $x$. A positive MI indicates that the fraction of neurons originating from *Frmd7tm* mice is decreased in the grid, while a negative MI indicates that the fraction is increased.

**Virus injections for retinal two-photon calcium imaging**. For intravitreal viral injections, mice were anesthetized with an intraperitoneal injection of a fentanyl (0.05 mg/kg body weight; Hameln), midazolam (5.0 mg/kg body weight; Hameln), and medetomidine (0.5 mg/kg body weight; Domitor, Orion) mixture dissolved in saline. We made a small hole at the border between the sclera and the cornea with a 30-gauge needle. Next, we loaded 2 μl of AAV1-CAG-GCaMP6s-WPRE-SV40

($1 \times 10^{13}$ vg/ml, Penn Vector Core, #AV-1-PV2833) into a pulled borosilicate glass micropipette, and the AAV was pressure-injected through the hole into the vitreous of the left eye using a Picospritzer III (Parker). Mice were returned to their home cage after anesthesia was reversed by an intraperitoneal injection of a flumazenil (0.5 mg/kg body weight; Hameln) and atipamezole (2.5 mg/kg body weight; Antisedan, Orion Pharma) mixture dissolved in saline.

**Retinal two-photon calcium imaging**. Retinal imaging was performed 3−4-weeks after virus injections. Mice were first dark-adapted for 1 h, and next the retina was prepared[68]. The retina was isolated from the left eye, and mounted ganglion-cell-side up on a small piece of filter paper (Millipore, MF-membrane), in which a 2 × 2 mm aperture window had previously been cut. During the procedure, the retina was illuminated by dim red light (KL1600 LED, Schott) filtered with a 650 nm high-pass optical filter (650/45×, Chroma) and bathed in extracellular solution (in mM): 110 NaCl, 2.5 KCl, 1 CaCl$_2$, 1.6 MgCl$_2$, 10 D-glucose, 22 NaHCO$_3$ bubbled with 5% CO$_2$, 95% O$_2$. The retina was kept at 35−36 °C and continuously superfused with oxygenated extracellular solution during recordings. For retinal two-photon calcium imaging we employed an equipment setup similar to that previously employed[68]. The isolated retina was placed under a microscope (SliceScope, Scientifica) equipped with a galvo-galvo scanning mirror system (8315 K, Cambridge Technologies), a mode-locked Ti:Sapphire laser tuned to 940 nm (MaiTai DeepSee, Spectra-Physics), and an Olympus 20× (1.0 NA) objective. The GCaMP6s signals emitted were passed through a set of optical filters (ET525/50 m, Chroma; lp GG495, Schott) and collected using a GaAsP detector (Scientifica). Images were acquired at 6−10 Hz using custom-made software developed by Zoltan Raics (SELS Software).

**Visual stimuli for retinal two-photon calcium imaging**. The visual stimulation was generated via custom-made software (Python and LabVIEW) developed by Zoltan Raics, projected by a DLP projector (LightCrafter Fiber E4500 MKII, EKB Technologies) coupled via a liquid light guide to an LED source (4-wavelength high-power LED Source, Thorlabs) with a 400 nm LED (LZ4-00UA00, LED Engin) through a band-pass optical filter (ET405/40×, Chroma), and focused onto the photoreceptor layer of the mounted retina through a condenser (WI-DICD, Olympus). The stimuli were exclusively presented during the fly-back period of the horizontal scanning mirror[68]. To measure directional tuning and TF preference, we presented 100% contrast black and white sinusoidal drifting gratings (mean intensity, 0.058 mW/cm$^2$). Light intensity was measured using a power meter (PM200, Thorlabs) and a spectrometer (USB4000-XR1, Ocean Optics). Drifting gratings were presented in 3 trials for 3 s at a time, with 3 s of gray screen between presentations, and shown in 8 different directions (0°, 45°, 90°, 135°, 180°, 225°, 270°, and 315°) in a pseudorandomized fashion, with a spatial frequency of 0.03 cycles/° and TFs of 0.3, 0.75, 1.2, and 1.8 Hz. To measure ON and OFF responses, we presented static flash spots (2 s in duration, 50, 100, 200, 400, 800 μm in diameter). To classify retinal cells into ON-OFF and non-ON-OFF populations, we used an ON-OFF index (OOI)[25] (Supplementary Fig. 9):

$$\text{OOI} = \frac{R_{\text{ON}} - R_{\text{OFF}}}{R_{\text{ON}} + R_{\text{OFF}}}$$

Where $R_{\text{ON}}$ and $R_{\text{OFF}}$ denote peak calcium responses during the static spot illumination phase, and the phase after the illumination, respectively. If the mean OOI for 50−800 μm spots was <0.3, the cell was defined as an ON-OFF cell. We calculated the DSI for individual cells, and defined cells with DSI > 0.3 as DS cells, similar to experiments performed in the cortex.

**Image analysis for retinal two-photon calcium imaging**. The raw two-photon scanning imaging data acquired was initially loaded into Matlab and converted into accessible image files. The ROIs for cell bodies of retinal cells were drawn in Matlab by fitting polygons, and selected based on mean and maximum fluorescence images. Fluorescence time courses were computed as the mean of all pixels within the ROI at each timepoint and were extracted in Matlab. The raw GCaMP6 fluorescence signals for each ROI were normalized ($\Delta F/F_0$) using the mean fluorescence ($F_0$) in a 2 s window before visual stimulation, and then synchronized with visual stimulus information. The $\Delta F/F_0$ signals were resampled using the *interp* function in Matlab, and smoothed by a moving average filter (width: two data points). To evaluate cell responsiveness, we determined a threshold for each cell as mean$_{\Delta F/F0}$ + 2 SD$_{\Delta F/F0}$, and any cell with response amplitudes higher than their threshold was defined as visually responsive and included in further analysis.

**Histology and confocal imaging**. To validate the injection site in the dLGN, mice were anesthetized with an intraperitoneal injection of a fentanyl (0.05 mg/kg body weight; Hameln), midazolam (5.0 mg/kg body weight; Hameln) and medetomidine (0.5 mg/kg body weight; Domitor, Orion) mixture dissolved in saline, and transcardially perfused with PBS and then with 4% paraformaldehyde (PFA). Brains were removed, fixed overnight in PFA and then transferred to PBS and stored at 4 °C. Brain slices (200 μm thick) were collected in the coronal plane using a vibratome (Leica, VT1000S). Slices were counterstained with 4′,6-diamidino-2-phenylindole (DAPI; 1:1000 dilution, ThermoFisher) before mounting with mounting medium (Fisher Scientific). Images of 1024 × 1024 pixels were acquired

using a confocal microscope (Zeiss LSM 780) with a 10× (0.45 NA) objective. To validate the specificity of starburst amacrine cell ablation in diphtheria toxin-injected *ChAT-Cre × LSL-DTR* mice and PBS-injected littermates, we performed immunohistochemical analyses of the retinas[27]. A nasal mark was applied to the eyes and fixed for 20 min at room temperature (RT) in 4% PFA before dissection. Afterwards, eyes were rinsed in PBS, dissected, mounted on flatmount paper in 4% PFA for 30 min at RT and then washed with PBS overnight at 4 °C on a shaker. The next day, retinas were incubated in 30% sucrose in PBS for at least 3 h at RT. Afterwards, retinas were transferred in the sucrose buffer to microscope slides (SUPERFROST PLUS, Thermo Scientific) and frozen and thawed three times using dry ice to enhance antibody penetration. After washing with PBS, retinas were blocked for 3 h in blocking buffer (1% bovine serum albumin (BSA), 10% normal donkey serum (NDS), 0.5% TritonX 100, 0.02% sodium azide in PBS) at RT. Primary antibodies (rabbit anti-RBPMS 1:500 [Milipore, ABN1362] and goat anti-ChAT 1:200 [Milipore, ABN1144P]) were incubated for 5 days in antibody reaction buffer (1% BSA, 3% NDS, 0.5% TritonX 100, 0.02% sodium azide in PBS) at 4 °C on a shaker. Secondary antibodies (donkey anti-rabbit IgG Alexa Fluor 568 1:200 [Invitrogen], donkey anti-goat IgG Alexa Fluor 488 1:200 [Life Technologies] together with DAPI 1:1000 [ThermoFisher]) were incubated overnight at 4 °C in antibody reaction buffer. After a final wash in PBS, retinas were embedded in Fluoromount-G (eBioscience). For cell density analysis, z-stacks containing images of 1024 × 1024 pixels (1.38 μm per pixel) were acquired at an interval of 4 μm (total thickness of 75−80 μm) with a confocal microscope (Zeiss LSM 780) using a 10× (0.45 NA) objective, and cells were counted using ImageJ. For detailed confocal z-stacks, we used a 40× (1.4 NA) objective and acquired 1024 × 1024 pixel (0.35 μm per pixel) images at an interval of 0.3 μm (total thickness of ~75 μm).

**Statistical analysis**. Statistical tests were performed in Matlab and we used the following statistical tests where appropriate: Mann–Whitney *U*-test, Wilcoxon signed-rank, Kolmogorov–Smirnov, and $\chi^2$ with Yates correction. Rayleigh's test for non-uniformity of circular data was performed using the Circular Statistics Toolbox[69]. No testing was performed to check for normality or homogeneity of variance. Center and spread values are reported as mean ± SEM. We used no statistical methods to plan sample sizes, but used sample sizes similar to those frequently used in the field[6,56]. The number of animals and cells is included in the text or in figure legends. We did not use any randomization; data collection and analysis were not performed blind to the conditions of the experiments. No collected data were excluded from analysis. *P*-values <0.05 were considered to be statistically significant. When a statistical test was used, the *P*-value is noted either in the manuscript text or depicted in figures and legends as: *$P < 0.05$, **$P < 0.01$, ***$P < 0.001$, n.s., not significant, $P \geq 0.05$.

**Reporting summary**. Further information on research design is available in the Nature Research Reporting Summary linked to this article.

## Data availability

All relevant data is provided in the Source Data file or is available from the corresponding author upon reasonable request.

## Code availability

All relevant code is available from the corresponding author upon reasonable request.

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

## Acknowledgements

We thank Ashley Juavinett and Kachi Odoemene for generously helping us to implement ISOI, Daniel E. Wilson, Georg Keller, and Felix Widmer for technical advice on setting up in-vivo two-photon calcium imaging, Botond Roska for *Frmd7tm1a(KOMP)Wtsi* and *Chattm2(cre)Lowl*/MwarJ mice, Zoltan Raics for developing our visual stimulation system, and Bjarke Thomsen, Misugi Yonehara, Szilard Sajgo, and Stella Solveig Nolte for technical assistance. We also thank Alexander Attinger, Karl Farrow, Kenta M. Hagihara, and Sara Oakeley for critical comments on the manuscript. We acknowledge the following grants: Lundbeck Foundation PhD Scholarship (R230-2016-2326) to R.R., VELUX FONDEN Postdoctoral Ophthalmology Research Fellowship (27786) to A.M., Lundbeck Foundation PhD Scholarship (R249-2017-1614) to M.D.S., Lundbeck Foundation (DANDRITE-R248-2016-2518; R252-2017-1060), Novo Nordisk Foundation (NNF15OC0017252), Carlsberg Foundation (CF17-0085), and European Research Council Starting (638730) grants to K.Y.

## Author contributions

R.R., A.M. and K.Y. conceived and designed the experiments and analyses. R.R. performed the intrinsic signal optical imaging experiments; R.R. and A.M. analyzed the data. R.R. performed the in-vivo calcium imaging experiments; R.R. and A.M. analyzed the data. A.M. performed the in-vitro calcium imaging experiments; R.R. and A.M. analyzed the data. M.S.D. performed the optomotor reflex experiments; M.S.D. and R.R. analyzed the data. R.R., M.S.D. and K.Y. performed the immunohistochemistry experiments; R.R., M.S.D. and K.Y. analyzed the data. R.R., A.M. and K.Y. interpreted the data and wrote the paper.

## Competing interests

The authors declare no competing interests.
