## [Peer Review File · Nature Communications]

Reviewers' Comments:

Reviewer #1:

Remarks to the Author:

A very important paper exploring the relationship between retinal cell types and cortical function. The relationship between retinal direction selective neurons and cortical direction selectivity has been unclear despite evidence in mice and marmosets for DS neurons in retina and in thalamus and V1 (Layers 2-3). Here the authors used an elegant collection of KO mice and imaging to identify the cortical area (RL) that relies most on retinal DS to generate its own DS responses of certain speed and magnitude.

This is a remarkably thorough and clear paper. One serious issue stands out however:

The mutant alters (as I understand), ON-DS cells and while there are ON-DS responses in mouse dLGN, the ON-DS cells altered in those mice do not project to dLGN but rather, to accessory optic targets MTN, LTN, etc- a system the senior author is very familiar with and indeed pioneered the genetic analysis of in Noda and Roska labs.

So I am perplexed: how does the retinal (lack of) information in retina of these mutants, translate to lack of specific DS responses in RL? The information those cells carry never makes it to V1 or cortex at all.

Please clarify. Is it all due to starburst amacrine deficiencies? What is the evidence that ON-OFF DSGCs (which do project to dLGN) are disrupted.

Again, a great question and format of analysis/inquiry is spelled out here but this needs clarifying and a schematic in the final figure that includes cell types in the retino-dLGN-V1(L1-3) > RL pathway.

Reviewer #2:

Remarks to the Author:

Rasmussen et al reports a cortical stream specialized for high speed motion selectivity. The study employed Frmd7 mutant mice in which retinal direction selectivity is disrupted. In these mice, the horizontal direction selectivity was impaired in LGN axons in L1/2 of V1, neurons in L2/3 of V1 and in area RL, but not in neurons in L4 of V1. The study proposes a segregated pathway that conveys the retinal direction selectivity to the LGN axons in L1, L2/3 in V1, and to the area RL, bypassing the L4 in V1.

This study investigates an important topic in visual neuroscience, how the retinal motion computation contributes to the cortical processing. Although some of the pathways reported in this study have been reported by other groups, this study investigated the cortical pathways more thoroughly, and identified specific contribution of retinal direction selectivity to the L2/3 of V1 and to RL bypassing L4. While this finding is novel and exciting, the manuscript needs some improvements with additional experiments to be published in Nature Communications.

First, this study investigates visual processing in higher visual areas under anesthesia. In general, response in higher sensory areas is severely attenuated by anesthesia (much so compared to primary sensory cortex and thalamus). Studies have shown that even in primary sensory cortex, the response amplitude under anesthesia is significantly attenuated (Niell and Stryker 2010) and the response pattern could be drastically affected due to weaker inhibition (Haider et al 2013). Therefore, considering the current standard of research in visual neuroscience, findings on higher sensory areas under anesthesia will cause more reservation than it used to.

Second, the study uses mutant mice to investigate the specific pathway. The effect of genetic mutant on cortical circuits is chronic, which can trigger many kinds of neural plasticity during development. Such plasticity leads to complications in clarifying specific pathways for physiological sensory processing. If this study can strengthen their findings with acute ablation/suppression experiments, the conclusions of this study will draw greater general interests, maybe even under anesthesia. It is worth noting that Hillier et al 2017 (cited in this paper) used the same mutant mice, but their findings are mainly based on separate ablation experiments.

Other major concerns are the following:

First, the study categorizes the response of visual neurons based on unsupervised clustering. It is an interesting approach, but the distance among the clusters looks very small, and it is not clear how robust and reproducible these clusters are. Thus, the validity of this approach needs to be shown. For example, does the grouping remain the same with different clustering methods? This study used different calcium sensors in different experiments, could it affect the clustering? Does the clusters remain the same across different brain states (awake vs. anesthetized). Alternatively, in the reviewer's view, the plots like Fig. 2d look much more convincing than this clustering approach, so eliminating the clustering descriptions is a possibility.

Second, the reviewer is puzzled by the temporal-frequency dependent characteristics in wild-type mice. Fig. 2d (RL) and Fig. 4c (V1 L2/3) look extremely similar to each other (a separate cluster in N direction), but not to Fig. 6j (thalamic butons in L1/2) and Fig. 7h (retina). This may indicate the actual processing is more complicated than what this study proposes.

Minor comments

1) The study often uses the term 'speed' to describe the response of RL neurons. This term is confusing, because the study only changes the temporal frequency of drifting gratings without changing spatial frequency (except Fig. S4). To investigate speed, various combinations of spatial and temporal frequency need to be explored (Andermann et al 2011). Their conclusions do not depend on the usage of this term 'speed'. The reviewer strongly recommends that the term 'speed' should be replaced with some other term.

2) The study should state that their RL may include area A, the existence of which is still debated in the field. In particular, this study uses intrinsic signal under anesthesia, and this signal is considered to be weak in higher visual areas. (That is why Marshel et al 2011 used wide-field calcium imaging and Andermann et al 2011 used autofluorescence imaging under awake conditions to identify the locations of the higher visual areas.)

3) The third section of the Result: The study uses retrograde viral GCaMP6m expression (rAAV) to explore the response patterns of the projection neurons. On the other hand, this study investigated the control response as target-unspecific neurons using GCaMP6f. The study should have used GCaMP6m for target-unspecific neurons. Readers need to know why comparing GCaMP6f and GCaMP6m data is valid.

4) Does the number of RL projecting neurons remain similar in Frmd7tm?

5) Discussion on Area RL: Area RL is also known to be involved in visuo-motor integration (Itokazu et al 2018) and in visual navigation (Harvey and Tank 2012, Krumin et al 2018). Moreover, please consider referring to de Vries et al (BioRxive) which reports direction selectivity in V1 and RL.

6) Fig. 2d: Please consider showing the same plot for Frmd7tm. Such a figure should be more important than dozens of plots shown in Fig. 2bc

Reviewer #3:

Remarks to the Author:

The manuscript by Rune and colleagues makes several important discoveries about the routing of retinal direction-selective responses to higher visual areas in mouse cortex. This manuscript will be of significant interest to vision scientists and the large community of neuroscientists working to understand the processing and routing of signals through mouse cortex. The study utilizes a recently introduced knockout line (Frmd7tm) in which the responses of direction-selective retinal ganglion cells that prefer motion along the horizontal axis (posterior or anterior) are disrupted: direction-selective cells preferring motion along the vertical axis are preserved (superior and inferior). The authors find that in the Frmd7 mice, many fewer neurons in area RL (but not PM) fail to generate robust responses to fast (40 deg/s) posterior motion. Furthermore, these signals are at least partly routed through layer 2/3 of V1. This routing is somewhat specific, as signals are relatively unperturbed in the Frmd7 mice in layer 4 of V1 and in area PM. In sum this study adds significantly to our understanding of how direction-selective signals that originate in the retina are routed through the LGN, V1, and area RL. The writing is clear and the figures are beautifully organized.

I have one significant concern, which is the clustering approach that is used by the authors throughout to the manuscript (Figs 2,4 and 5) and upon which many of their results are based. The authors utilize t-SNE to generate a low dimension representation of their data upon which they cluster. However, this approach is prone to a number of serious problems that could cause the authors to misunderstand their data. A summary of these problems can be found here:

<https://stats.stackexchange.com/questions/263539/clustering-on-the-output-of-t-sne/264647>. In short, t-SNE is a visualization tool, not a clustering tool. While it has been used for clustering — particularly in biology — this application is at best controversial, and at worst an error that produces specious clustering. The main problem arises because t-SNE does not preserve distance relationships in data — points that are close in the original data can be far from one another after t-SNE. Thus, the authors must cluster their data using a different approach — ideally one that preserves relative distance relationships. The dimensionality of the data is sufficiently low and the authors have collected data from a lot of neurons, so I anticipate this will be achievable. For example, using PCA to reduce the dimensionality of the data and then performing hierarchical clustering or k-means clustering should be tested. I suspect that the results obtained by the authors will survive this more traditional clustering/analysis approach, but it must be checked.

Minor concerns:

It was not clear to me from the text in the methods how clusters were matched across WT and Frmd2 mice. There are offsets in the responses across genotypes (particularly in speed, e.g. Fig 2) that seem significant and make me wonder whether there are additional differences in DS/speed tuning that are not being extracted by this analysis.

It would be useful to readers to see how the authors identified ON-OFF DSGCs from their GCAMP6s measurements. This isn't clear in Fig 7.

Why would the abundance of posterior tuned cells in the retina become clearer at higher speeds?

It is unclear the level of specificity in the connections between L1-3 of V1 and RL. It is clear that there is a relationship for responses to high speeds along the horizontal axis, but these cells are responding to many other stimuli as well — so describing these connections as 'highly-specific' (line 285), seems not well supported.

It would be helpful to readers to say more about the specificity of the Frmd7 line in the Discussion. How confident are we that the perturbations in this line are retina-specific? Are there any caveats to the interpretation that all the effects observed in HVAs result from changes in the retina?

Point-by-point list of responses to the reviewers' comments

We genuinely thank the three reviewers for their valuable comments on our manuscript. We have strived to make all the requested improvements where possible, in order to strengthen the manuscript. Our comments are written below the original comments made by reviewers 1, 2, and 3.

Reviewer #1 (Remarks to the Author):

A very important paper exploring the relationship between retinal cell types and cortical function. The relationship between retinal direction selective neurons and cortical direction selectivity has been unclear despite evidence in mice and marmosets for DS neurons in retina and in thalamus and V1 (Layers 2-3). Here the authors used an elegant collection of KO mice and imaging to identify the cortical area (RL) that relies most on retinal DS to generate its own DS responses of certain speed and magnitude.

This is a remarkably thorough and clear paper. One serious issue stands out however:

The mutant alters (as I understand), ON-DS cells and while there are ON-DS responses in mouse dLGN, the ON-DS cells altered in those mice do not project to dLGN but rather, to accessory optic targets MTN, LTN, etc. a system the senior author is very familiar with and indeed pioneered the genetic analysis of in Noda and Roska labs. So I am perplexed: how does the retinal (lack of) information in retina of these mutants, translate to lack of specific DS responses in RL? The information those cells carry never makes it to V1 or cortex at all. Please clarify. Is it all due to starburst amacrine deficiencies? What is the evidence that ON-OFF DSGCs (which do project to dLGN) are disrupted?

We thank the reviewer for the thorough and insightful comments on our manuscript. The reviewer raises an important point regarding how the retinal *Frmd7* mutation affects specific types of retinal DS cells, and thereby also the projection targets downstream of the retina. Importantly, both ON-DS and ON-OFF DS cells are altered in *Frmd7tm* mice as demonstrated previously: ON-OFF DS cells were identified by multi-electrode array recordings and two-photon-targeted patch-clamp recordings from genetically-labeled (either Hoxd10-EGFP- or *Drd4*-EGFP-labeled) ON-OFF DS cells (Yonehara et al., *Neuron* 2016; Hillier et al., *Nat Neurosci* 2017). In the revised manuscript we have added new imaging experiments from mice in which starburst amacrine cells are genetically ablated from the retina; it is known that mouse retinas in which starburst amacrine cells are ablated lose retinal ON-OFF and ON direction selectivity (Yoshida et al., *Neuron* 2001; Hillier et al., *Nat Neurosci* 2017). As the reviewer points out, it is unclear whether ON-DS cells, which project exclusively to accessory optic targets (Yonehara et al., *PLoS One* 2008, 2009; Dhande et al., *J Neurosci* 2013), may affect the responses of RL neurons via accessory optic targets. Another strong argument for why we propose that the disruption of ON-DS cells unlikely translate to lack of specific DS responses in area RL is the slow speed tuning of ON-DS cells; ON-DS cells respond best to motion at speeds of around 5°/s (corresponding to the temporal frequency of 0.15Hz in this manuscript) and their firing significantly decreases at higher speeds (Yonehara et al., *Neuron* 2016; Matsumoto et al., *Curr Biol* 2019), whereas ON-OFF DS cell's firing steadily increases as the stimulus speed increases until around 20-40°/s (Weng et al., *J Physiol* 2005; Yonehara et al., *Neuron* 2016; Fig. 7d in this manuscript). We observed that the lack of specific DS responses in area RL is observed at the speed of around 40°/s (1.2Hz in this manuscript) but not

at 10°/s (0.3Hz). Therefore, we propose that at least the major effect on DS responses in area RL of *Frmd7tm* mice is mediated by altered responses of retinal ON-OFF DS cells projecting to the dLGN.

Again, a great question and format of analysis/inquiry is spelled out here but this needs clarifying and a schematic in the final figure that includes cell types in the retino-dLGN-V1 (L1-3) > RL pathway.

Thank you for pointing out this issue. In the schematic of Figure 7j in the revised manuscript we have specified that it is the retinal ON-OFF DS cells that are affected in the retino-thalamo-cortical pathway proposed. The updated schematic and its corresponding legend are shown below:

Fig. 7: Preference of retinal neurons to posterior motion at higher TFs is disrupted in *Frmd7tm* mice...
 (j) Schematic diagram of proposed neural pathway linking retinal ON-OFF DS cells to RL DS cells.

Reviewer #2 (Remarks to the Author):

Rasmussen et al reports a cortical stream specialized for high speed motion selectivity. The study employed *Frmd7* mutant mice in which retinal direction selectivity is disrupted. In these mice, the horizontal direction selectivity was impaired in LGN axons in L1/2 of V1, neurons in L2/3 of V1 and in area RL, but not in neurons in L4 of V1. The study proposes a segregated pathway that conveys the retinal direction selectivity to the LGN axons in L1, L2/3 in V1, and to the area RL, bypassing the L4 in V1. This study investigates an important topic in visual neuroscience, how the retinal motion computation contributes to the cortical processing. Although some of the pathways reported in this study have been reported by other groups, this study investigated the cortical pathways more thoroughly, and identified specific contribution of retinal direction selectivity to the L2/3 of V1 and to RL bypassing L4. While this finding is novel and exciting, the manuscript needs some improvements with additional experiments to be published in Nature Communications.

First, this study investigates visual processing in higher visual areas under anesthesia. In general, response in higher sensory areas is severely attenuated by anesthesia (much so compared to primary sensory cortex and thalamus). Studies have shown that even in primary sensory cortex, the response amplitude under anesthesia is significantly attenuated (Niell and Stryker 2010) and the response pattern could be drastically affected due to weaker inhibition (Haider et al 2013). Therefore, considering the current standard of research in visual neuroscience, findings on higher sensory areas under anesthesia will cause more reservation than it used to.

We thank the reviewer for his/her thorough and insightful comments on our initial manuscript. The reviewer raises an important point, and we fully agree that it is important to validate and assess our findings in higher visual areas in awake mice. We have therefore carried out additional experiments in awake, quietly resting mice: we performed two-photon calcium imaging from L2/3 of areas RL and PM in control and *Frmd7tm* mice. To habituate the mice to the experimental conditions, each mouse was head-fixed onto the imaging stage, with its body restrained in a cylindrical cover. The habituation procedure was repeated for 3 days for each mouse at durations of 15, 30 and 60 mins. In short, these experiments showed that the key characteristic findings in areas RL and PM of anesthetized mice were preserved in awake mice. RL DS cells in awake control mice preferred fast motion moving along the horizontal axis, while PM DS cells preferred slower motion (i.e. low temporal frequency [TF]) equally across all directions. In awake *Frmd7tm* mice, RL DS cells preferred slower motion, whereas the preference of PM DS cells was not affected. Importantly, as the TF increased from 0.3 to 1.2Hz, RL DS cells in awake control mice developed a posterior motion bias, and this bias was significantly affected in *Frmd7tm* mice. In area PM, preferred motion direction distributions did not differ between awake control and *Frmd7tm* mice at neither 0.3 nor 1.2Hz. The supplementary figure summarizing these results is shown below, together with its corresponding legend.

Supplementary Fig. 3: Preference of RL neurons for posterior motion at higher TFs depends on retinal horizontal direction selectivity in awake mice. (a) Two-photon calcium imaging was performed from L2/3 in areas RL and PM of awake control mice (1,652 and 2,018 DS cells, respectively; 4 mice) and *Frmd7tm* mice (2,093 and 4,049 DS cells, respectively; 3 mice). Example image shows two-photon mean projection image of RL neurons expressing GCaMP6f (scale bar, 100 μm). Example traces show activity from three neurons (circled in the image) recorded while the mouse was awake and quietly resting in the cylindrical cover. (b) Fraction of DS cells in RL and PM ($P \geq 0.05$ for both comparisons, χ^2 test with Yates correction). (c) Preferred TF for DS cells in RL ($P < 0.001$, Mann-Whitney U test) and PM ($P \geq 0.05$, Mann-Whitney U test). Triangles show medians. (d) Response amplitude as a function of motion direction and TF for RL and PM DS cells. White and black asterisks: significantly decreased and increased response amplitude in *Frmd7tm* mice, respectively, Mann-Whitney U test. (e) Fractional distributions of preferred motion directions for RL and PM DS cells at 0.3 and 1.2Hz; fractions are normalized to the largest fraction across genetic groups. (f) Distributions of preferred direction at 0.3 and 1.2Hz in RL and PM. *** $P < 0.001$, Kolmogorov-Smirnov test. Source data are provided as a Source Data file.

Second, the study uses mutant mice to investigate the specific pathway. The effect of genetic mutant on cortical circuits is chronic, which can trigger many kinds of neural plasticity during development. Such plasticity leads to complications in clarifying specific pathways for physiological sensory processing. If this study can strengthen their findings with acute ablation/suppression experiments, the conclusions of this study will draw greater general interests, maybe even under anesthesia. It is worth noting that Hillier et al 2017 (cited in this paper) used the same mutant mice, but their findings are mainly based on separate ablation experiments.

Thank you for bringing this point to our attention. We fully agree that this is an important point that should be investigated experimentally. In the revised manuscript we have carried out ablation experiments: we specifically ablated starburst amacrine cells in adult mice by genetically targeting the diphtheria toxin receptor to ChAT-positive cells and intravitreally injecting diphtheria toxin. Using these starburst-ablated mice, in which retinal direction selectivity is disrupted (Yoshida et al., Neuron 2001; Hillier et al., Nat Neurosci 2017), we then assessed the downstream cortical affect by performing two-photon calcium imaging from L2/3 of areas RL and PM in awake mice. Injecting diphtheria toxin into the eyes of *ChAT-Cre* × *LSL-DTR* mice led to the selective loss of starburst amacrine cells, as shown by immunohistochemistry, and the loss of optomotor responses in all mice. From the two-photon imaging we found no differences in the fraction of DS cells between control and starburst-ablated mice. In starburst-ablated mice, RL DS cells preferred motion with a lower TF compared to control mice, while we found no difference for PM DS cells. Notably, RL DS cells from starburst-ablated mice lacked the response preference for horizontal motion at higher TFs and the posterior motion bias at 1.2Hz was significantly impaired, whereas we found no difference in motion bias for PM DS cells. The supplementary figures summarizing these results are shown below, together with their corresponding legends.

Supplementary Fig. 4: Diphtheria toxin injection selectively ablates starburst amacrine cells in the retina and annihilates optomotor responses. (a) Whole-mount retinas stained for ChAT to label starburst amacrine cells (white dots show somata) in retinas from control (PBS-injected) and starburst-ablated mice (diphtheria toxin-injected). The white dots in the starburst-ablated retina are fluorescence aggregates, not somata (scale bar, 1 mm). (b) Higher magnification of the retinas shown in (a) showing the absence of starburst amacrine cells in ablated mice (scale bar, 250 μm). (c) Side view (top; scale bar, 30 μm) and top view (bottom; scale bar, 30 μm) of retinal z-projection stained for ChAT (starburst amacrine cells) from control (left) and starburst-ablated (right) mice. (d) Side view (top; scale bar, 30 μm) and top view (bottom; scale bar, 30 μm) of retinal z-projection stained for RBPM5 (retinal ganglion cells) from control (left) and starburst-ablated (right) mice. GCL, ganglion cell layer. INL, inner nuclear layer; IPL, inner plexiform layer. (e) Density quantification of starburst amacrine cells and retinal ganglion cells in control and starburst-ablated mice (7 retinas in each group; non-significant [n.s.], $P \geq 0.05$, *** $P < 0.001$, Mann-Whitney U test). Circles are individual data points, center line is median, box limits are 25th and 75th percentiles, and whiskers show minimum and maximum values. (f) Horizontal optomotor response measured in control (5 mice, 7–8 trials per mouse) and starburst-ablated mice (5 mice, 8 trials per mouse). The dotted horizontal line represents the upper quartile of the OMR index previously collected from 3 blind control mice (*rd1/rd1* mutants)¹. Shading indicates SEM.

Supplementary Fig. 5: Ablating retinal starburst cells impairs posterior motion preference of RL DS cells at higher TFs. (a) Two-photon calcium imaging was performed from L2/3 in areas RL and PM of awake control mice (1,652 and 2,018 DS cells, respectively; 4 mice) and starburst-cell-ablated mice (2,511 and 2,777 DS cells, respectively; 4 mice). (b) Fraction of DS cells in RL and PM ($P \geq 0.05$ for both comparisons, χ^2 test with Yates correction). (c) Preferred TF for DS cells in RL ($P < 0.001$, Mann-Whitney U test) and PM ($P \geq 0.05$, Mann-Whitney U test). Triangles show medians. (d) Response amplitude as a function of motion direction and TF for RL and PM DS cells. White and black asterisks: significantly decreased and increased response amplitude in starburst-ablated mice, respectively, Mann-Whitney U test. (e) Fractional distributions of preferred motion directions for RL and PM DS cells at 0.3 and 1.2Hz; fractions

are normalized to the largest fraction across genetic groups. (f) Distributions of preferred direction at 0.3 and 1.2Hz in RL and PM. *** $P < 0.001$, Kolmogorov-Smirnov test. Source data are provided as a Source Data file.

Other major concerns are the following:

First, the study categorizes the response of visual neurons based on unsupervised clustering. It is an interesting approach, but the distance among the clusters looks very small, and it is not clear how robust and reproducible these clusters are. Thus, the validity of this approach needs to be shown. For example, does the grouping remain the same with different clustering methods? This study used different calcium sensors in different experiments, could it affect the clustering? Does the clusters remain the same across different brain states (awake vs. anesthetized). Alternatively, in the reviewer's view, the plots like Fig. 2d look much more convincing than this clustering approach, so eliminating the clustering descriptions is a possibility.

We agree with the potential problems in the clustering methods using t-SNE and k-means methods to characterize and dissect the affected response properties in *Frmd7tm* mice, and such analyses have confused the purpose of our analyses; the main purpose of these analyses is not to understand how many functional neuronal groups exist in the relevant cortical areas, but instead to capture a trend in the response properties that are altered in *Frmd7tm* mice. We have thus removed these analyses from the revised manuscript. Instead, we have implemented a simpler analytical procedure, which does not involve clustering, to determine the trends in affected response properties in cortical neurons from *Frmd7tm* mice. We first composed a TF-dependent response matrix for each visual area (e.g. area RL) using a total of eight parameters from each neuron: peak $\Delta F/F_0$ amplitudes and DSI at each of the four TFs (Fig. 2a). The response matrix included datasets from both control and *Frmd7tm* mice. Next, the response matrix was decomposed into two dimensions by standard principal component analysis (PCA). The resulting PCA distributions revealed a clear distribution trend depending on the TF preference of the neurons, indicating that neurons sharing the same TF preference tended to be localized close to each other in the PCA distribution (Fig. 2b, c).

We next segmented the PCA distribution into 8×8 grids to calculate the fraction of neurons located within each of the grids, and examined the fractional changes between control and *Frmd7tm* mice (Fig. 2d). Based on the fractional changes we statistically classified neurons into three groups: 1) "Increased", 2) "Decreased", and 3) "Unchanged" in *Frmd7tm* mice, compared to control mice (Fig. 2e). Afterwards we probed the functional characteristics of neurons from each of these three main groups; we analyzed TF-dependent peak response amplitudes (Fig. 2f), preferred directions, and DSI (Fig. 2g). This analysis enabled us to correlate the TF-dependent response properties in individual cortical DS cells and the fractional changes of neurons between control and *Frmd7tm* mice without relying on clustering algorithms that are prone to a number of serious problems that could lead to misunderstanding the data.

Regarding the potential effect of the uses of different calcium sensors on clustering, this concern should be less relevant now since we have removed clustering analyses from the revised manuscript. In our new analysis, we relied on simple response features (i.e. $\Delta F/F_0$ amplitudes and DSI) from each neuron to compose the response matrix decomposed using PCA. Previous work has shown that the relationship between peak GCaMP6 $\Delta F/F_0$ and the number of evoked action potentials is relatively linear within each of the GCaMP6 subtypes, and is very similar for GCaMP6f and GCaMP6m (Chen et al., Nature 2013). Importantly, we used

GCaMP6f and GCaMP6m either for 1) comparing responses between two genetic conditions within the same GCaMP6 subtype (Fig. 3), or comparing projection pattern-related response differences within the same GCaMP6 subtype (Fig. 4h). Therefore, we see no reason why this should impact our conclusion. The results of our updated analysis (shown here for area RL) are shown below, together with the corresponding legend.

Fig. 2: Neurons with distinct functional characteristics are sensitive to disruption of retinal horizontal direction selectivity in area RL. (a) Response matrix composed of TF-dependent response amplitudes and DSI for RL L2/3 DS cells sorted by TF preference in control and *Frmd7tm* mice. (b) 2D visualization of the 1st and 2nd principal components for the response matrix shown in (a). Each point represents one neuron. (c) TF preference of individual RL neurons in control and *Frmd7tm* mice. (d) Fraction of neurons in 8×8 grids (gray lines) calculated from the PCA plot shown in (b) for control and *Frmd7tm* mice. (e) Fraction difference map between control and *Frmd7tm* mice. Black and white asterisks: significantly decreased and increased fractions in *Frmd7tm* mice, respectively, $P < 0.05$, χ^2 test with Yates correction. (f) Peak response amplitude as a function of TF for three groups (decreased, increased, or unchanged in *Frmd7tm* mice) in control and *Frmd7tm* mice. (g) TF-dependent tuning characteristics of individual RL neurons from the three groups in control and *Frmd7tm* mice. Angular coordinate: preferred direction. Radial coordinate: DSI. Inner circle: DSI of 0.5.

Second, the reviewer is puzzled by the temporal-frequency dependent characteristics in wild-type mice. Fig. 2d (RL) and Fig. 4c (V1 L2/3) look extremely similar to each other (a separate cluster in N direction), but not to Fig. 6j (thalamic boutons in L1/2) and Fig. 7h (retina). This may indicate the actual processing is more complicated than what this study proposes.

This is an important observation and we agree that the actual processing should be more complicated than a simple model in which the activity of retinal ON-OFF DS cells linearly modulates the responses of V1 and RL L2/3 neurons. To respond to the reviewer's request that the t-SNE and unsupervised clustering should be reconsidered, we have removed these data from the revised manuscript. Nevertheless, here we would like to answer this issue based on data presented in our original manuscript. The appearance of a separate cluster in

Fig. 2d and Fig. 4c, but not in Fig. 6j and Fig. 7h, may suggest an intracortical processing mechanism that toggle the behavior of area RL and V1 L2/3 between two states abruptly across a certain threshold of TF, and recurrent excitatory circuits between V1 L2/3 and RL L2/3 cells, which would make the Fig. 2d (area RL) and Fig. 4c (V1 L2/3) look similar to each other. Here we propose two potential mechanisms. Firstly, the suppression switch of RL G1-5 and V1 G1-7 cells may become “on” in response to a stimulus with a lower temporal frequency than a certain threshold. One potential driver of such suppression is input from V1 L4. Secondly, the activation switch of the RL G1-5 and V1 G1-7 cells may become “on” in response to a stimulus with higher temporal frequency than a certain threshold, where the summed activity level of converging inputs from retina-dLGN posterior-preferring DS cells reaches a certain threshold. The first and second mechanisms may work simultaneously. These ideas are consistent with a cortical amplification model, which proposes that one of the key functions of the cortex is to amplify weakly biased inputs (Christie et al., J Neurophysiol 2017). Although what the exact mechanisms for the “circuit switch” are is an interesting question, we view this as outside the scope for this paper and would leave it as an exciting challenge for the future.

Minor comments

1) The study often uses the term ‘speed’ to describe the response of RL neurons. This term is confusing, because the study only changes the temporal frequency of drifting gratings without changing spatial frequency (except Fig. S4). To investigate speed, various combinations of spatial and temporal frequency need to be explored (Andermann et al 2011). Their conclusions do not depend on the usage of this term ‘speed’. The reviewer strongly recommends that the term ‘speed’ should be replaced with some other term.

We thank the reviewer for pointing out this issue regarding the confusion of the term “speed” that we used to describe RL neurons. We fully agree with the reviewer that it is more appropriate to use the term “temporal frequency” since we did not sufficiently explore the full visual stimulus space in order to confidently use the term “speed” in our description. In the revised manuscript we have therefore replaced all occurrences of the term “speed” with the term “temporal frequency” when describing the condition of a certain fixed spatial frequency.

2) The study should state that their RL may include area A, the existence of which is still debated in the field. In particular, this study uses intrinsic signal under anesthesia, and this signal is considered to be weak in higher visual areas. (That is why Marshel et al 2011 used wide-field calcium imaging and Andermann et al 2011 used autofluorescence imaging under awake conditions to identify the locations of the higher visual areas.)

This is an important point and we thank the reviewer for bringing this to our attention. In the revised manuscript we have therefore included the following text that highlights that area RL in our experiments might also include area A (page 3):

“It is worth noting that the RL area we identified may possibly include the anterior HVA (area A), which has been identified previously (Marshel et al. Neuron 2011; Andermann et al. Neuron 2011)”.

3) The third section of the Result: The study uses retrograde viral GCaMP6m expression (rAAV) to explore the response patterns of the projection neurons. On the other hand, this study investigated the control

response as target-unspecific neurons using GCaMP6f. The study should have used GCaMP6m for target-unspecific neurons. Readers need to know why comparing GCaMP6f and GCaMP6m data is valid.

We thank the reviewer for the comment. Actually, in the analyses included in this manuscript, we never directly compared data obtained with GCaMP6f and GCaMP6m: no statistical comparisons were performed between GCaMP6f and GCaMP6m data. Indeed, what we have compared is the responses between control and *Frmd7tm* mice by using the same GCaMP6 subtypes. In the manuscript, we do discuss the TF-dependent preferred motion bias between neurons imaged using GCaMP6f and GCaMP6m, but we think this should not be a problem assuming that the relationship between peak GCaMP6 $\Delta F/F_0$ and the number of evoked action potentials is relatively linear within each of the GCaMP6 subtypes, as suggested previously (Chen et al., Nature 2013). Although we pooled V1 L2/3 imaging data obtained using GCaMP6f and GCaMP6m to compare control and *Frmd7tm* mice, this should not affect the main conclusion of the analysis for the same reason as described above. That being said, we agree that using the same GCaMP subtype was the most careful approach that could have been chosen.

4) Does the number of RL projecting neurons remain similar in *Frmd7tm*?

To test this point, we determined the density of GCaMP6m-expressing neurons (somata) in each two-photon imaged field of view (FOV; $471 \times 471 \mu\text{m}$) from control and *Frmd7tm* mice. This analysis showed that the rough density of RL-projecting V1 neurons did not significantly differ between genetic conditions (68 ± 1.4 neurons/FOV and 81 ± 1.04 neurons/FOV in control and *Frmd7tm* mice, respectively, $n = 21$ FOVs, $P = 0.13$, Mann-Whitney U test). Similarly, we found no significant difference for PM-projecting V1 neurons (86 ± 0.97 neurons/FOV and 76 ± 1.02 neurons/FOV in control and *Frmd7tm* mice, respectively, $n = 36/34$ FOVs, $P = 0.18$, Mann-Whitney U test). We have added this result to the supplementary figure 7 and to the revised manuscript with the following text (page 6):

“The density of GCaMP6-labelled projection neurons did not differ significantly between genetic conditions (Supplementary Fig. 7e).”

5) Discussion on Area RL: Area RL is also known to be involved in visuo-motor integration (Itokazu et al 2018) and in visual navigation (Harvey and Tank 2012, Krumin et al 2018). Moreover, please consider referring to de Vries et al (BioRxiv) which reports direction selectivity in V1 and RL.

We thank the reviewer for referring us to these important references. We agree that they should be included in our discussion on area RL and we have therefore included Itokazu et al. 2018, Harvey and Tank 2012, and Krumin et al. 2018 in the revised manuscript. On the other hand, we have not found any persuasive reason to refer to de Vries et al (BioRxiv) in our manuscript.

6) Fig. 2d: Please consider showing the same plot for *Frmd7tm*. Such a figure should be more important than dozens of plots shown in Fig. 2bc.

We agree with the reviewer that also showing this information for *Frmd7tm* mice is important. In the revised manuscript we have updated Figure 2 so that it now shows our new decomposition and segmentation analysis and results. To encompass the suggestion made by the reviewer, we have included response characterization

plots for both control and *Frmd7tm* mice. The information shown in these plots includes TF-dependent peak response amplitude, TF-dependent DSI, and TF-dependent direction tuning. The updated panels for area RL from Figure 2 are shown here, together with its corresponding legend.

Fig. 2: Neurons with distinct functional characteristics are sensitive to disruption of retinal horizontal direction selectivity in area RL... (f) Peak response amplitude as a function of TF for three groups (decreased, increased, or unchanged in *Frmd7tm* mice) in control and *Frmd7tm* mice. (g) TF-dependent tuning characteristics of individual RL neurons from the three groups in control and *Frmd7tm* mice. Angular coordinate: preferred direction. Radial coordinate: DSI. Inner circle: DSI of 0.5.

Reviewer #3 (Remarks to the Author):

The manuscript by Rune and colleagues makes several important discoveries about the routing of retinal direction-selective responses to higher visual areas in mouse cortex. This manuscript will be of the processing and routing of signals through mouse cortex. The study utilizes a recently introduced knockout line (*Frmd7tm*) in which the responses of direction-selective retinal ganglion cells that prefer motion along the horizontal axis (posterior or anterior) are disrupted: direction-selective cells preferring motion along the vertical axis are preserved (superior and inferior). The authors find that in the *Frmd7* mice, many fewer neurons in area RL (but not PM) fail to generate robust responses to fast (40 deg/s) posterior motion. Furthermore, these signals are at least partly routed through layer 2/3 of V1. This routing is somewhat specific, as signals are relatively unperturbed in the *Frmd7* mice in layer 4 of V1 and in area PM. In sum this study adds significantly to our understanding of how direction-selective signals that originate in the retina are routed through the LGN, V1, and area RL. The writing is clear and the figures are beautifully organized.

I have one significant concern, which is the clustering approach that is used by the authors throughout to the manuscript (Figs 2,4 and 5) and upon which many of their results are based. The authors utilize t-SNE to generate a low dimension representation of their data upon which they cluster. However, this approach is prone to a number of serious problems that could cause the authors to misunderstand their data. A summary of these problems can be found here: <https://stats.stackexchange.com/questions/263539/clustering-on-the-output-of-t-sne/264647>. In short, t-SNE is a visualization tool, not a clustering tool. While it has been used for clustering — particularly in biology — this application is at best controversial, and at worst an error that produces specious clustering. The main problem arises because t-SNE does not preserve distance relationships in data — points that are close in the original data can be far from one another after t-SNE. Thus, the authors must cluster their data using a different approach — ideally one that preserves relative distance relationships. The dimensionality of the data is sufficiently low and the authors have collected data from a lot of neurons, so I anticipate this will be achievable. For example, using PCA to reduce the dimensionality of the data and then performing hierarchical clustering or k-means clustering should be tested.

I suspect that the results obtained by the authors will survive this more traditional clustering/analysis approach, but it must be checked.

We thank the reviewer for the thorough and important comments on our clustering procedure in the original manuscript, and we thank the reviewer for pointing us towards the resource describing the issues with t-SNE analysis. We acknowledge that our original clustering procedure was not without issues, and we have therefore removed it from the revised manuscript. As the reviewer rightfully points out, our data dimensionality was rather low and we have collected data from many neurons, so we instead opted for a simpler and more robust analytical procedure, not involving clustering, for determining trends in affected response properties in cortical neurons from *Frmd7tm* mice. Here we first composed a TF-dependent response matrix for each visual area (e.g. area RL) using a total of eight parameters from each neuron: peak $\Delta F/F_0$ amplitudes and DSI at each of the four TFs. The response matrix included datasets from both control and *Frmd7tm* mice (Fig. 2a). Next, the response matrix was decomposed into two dimensions by PCA (Fig. 2b). The resulting PCA distributions revealed a clear distribution trend depending on the TF preference of the neurons, indicating that neurons sharing the same TF preference tended to be localized close to each other in the PCA distribution (Fig. 2c). We next segmented the PCA distribution into 8×8 grids to calculate the fraction of neurons located within each of the grids, and examined the fractional changes between control and *Frmd7tm* mice (Fig. 2d). Based on these fractional changes we statistically classified neurons into three groups: 1) “Increased”, 2) “Decreased”, and 3) “Unchanged” in *Frmd7tm* mice, compared to control mice (Fig. 2e). Afterwards we probed the functional characteristics of neurons from each of these three main groups; we analyzed TF-dependent peak response amplitudes (Fig. 2f), preferred directions, and DSI (Fig. 2g). This analysis allowed us to correlate the TF-dependent response properties in individual cortical DS cells, and the fractional changes of neurons between control and *Frmd7tm* mice without relying on any clustering algorithms. The results of our updated analysis (shown here for area RL) are shown below, together with the corresponding legend.

Fig. 2: Neurons with distinct functional characteristics are sensitive to disruption of retinal horizontal direction selectivity in area RL. (a) Response matrix composed of TF-dependent response amplitudes and

DSI for RL L2/3 DS cells sorted by TF preference in control and *Frmd7tm* mice. (b) 2D visualization of the 1st and 2nd principal components for the response matrix shown in (a). Each point represents one neuron. (c) TF preference of individual RL neurons in control and *Frmd7tm* mice. (d) Fraction of neurons in 8×8 grids (gray lines) calculated from the PCA plot shown in (b) for control and *Frmd7tm* mice. (e) Fraction difference map between control and *Frmd7tm* mice. Black and white asterisks: significantly decreased and increased fractions in *Frmd7tm* mice, respectively, $P < 0.05$, χ^2 test with Yates correction. (f) Peak response amplitude as a function of TF for three groups (decreased, increased, or unchanged in *Frmd7tm* mice) in control and *Frmd7tm* mice. (g) TF-dependent tuning characteristics of individual RL neurons from the three groups in control and *Frmd7tm* mice. Angular coordinate: preferred direction. Radial coordinate: DSI. Inner circle: DSI of 0.5.

Minor concerns:

It was not clear to me from the text in the methods how clusters were matched across WT and *Frmd7* mice. There are offsets in the responses across genotypes (particularly in speed, e.g. Fig 2) that seem significant and make me wonder whether there are additional differences in DS/speed tuning that are not being extracted by this analysis.

We fully agree with the reviewer that this was one of the issues with our clustering procedure employed in the original manuscript. As described in the response above, we have in the revised manuscript instead opted for a simpler analytical approach using data decomposition and segmentation rather than unsupervised clustering. In the new analysis described in the revised manuscript, this issue of matching clusters across control and *Frmd7tm* mice is thus no longer applicable as no clustering was performed.

It would be useful to readers to see how the authors identified ON-OFF DSGCs from their GCAMP6s measurements. This isn't clear in Fig 7.

This is a useful comment, and we fully agree that showing how we identified retinal ON-OFF DS cells is important for the reader. In brief, we first separated the visually responsive retinal cells into two groups: ON-OFF and non-ON-OFF cells, based on an ON-OFF index (OOI, see Methods section of manuscript). We classified cells with an $OOI < 0.3$ as ON-OFF cells. Next, we defined ON-OFF DS cells as ON-OFF cells with a $DSI > 0.3$. The supplementary figure summarizing this procedure is shown below, together with the corresponding legend.

Supplementary Fig. 9: Classification of ON-OFF DS in the retina. (a) To identify ON-OFF DS cells, we first separated the visually responsive retinal cells into two groups: ON-OFF and non-ON-OFF cells, based on an ON-OFF index (OOI, see Methods), denoting the ratio of responses during and after a static flash stimulus (ON phase, orange area; OFF phase, blue area). If cells are responding to both the ON and OFF phase, the OOI value is low: we classified cells with an $OOI < 0.3$ as ON-OFF cells. Example ON-OFF, OFF, and ON cells are shown. (b) We defined ON-OFF DS cells as ON-OFF cells with a $DSI > 0.3$. Example ON-OFF DS cells and ON-OFF non-DS cells are shown.

Why would the abundance of posterior tuned cells in the retina become clearer at higher speeds?

In our two-photon calcium imaging experiments we identified posteriorly-tuned retinal ganglion cells not by genetic markers, but by the response properties of the imaged cells (we used an unspecific promoter, *CAG*, for expressing GCaMP6 in the retina). From the imaging data we identified ON-OFF DS cells follows. First we identified visually responsive cells by a threshold criterion, then we separated responsive cells into ON-OFF and non-ON-OFF cells, and then we calculated the DSI of the ON-OFF cells. If the ON-OFF cell showed a $DSI > 0.3$ at that particular TF (i.e. speed in original manuscript) then it was included for the preferred direction distribution analysis. Thus, since the responses of ON-OFF DS cells is sensitive to motion speed (Weng et al., *J Physiol* 2005; Yonehara et al. *Neuron* 2016), some ON-OFF cells would only be included in the preferred direction distribution in a subset of TFs since their DSI was not > 0.3 at all TFs. In this way, the abundance (or fraction in our analysis) of posteriorly-tuned ON-OFF DS cells could increase at higher speeds due to their preferential increase in peak response amplitude and DSI as function of motion speed, compared to ON-OFF DS cells preferring other motion directions.

It is unclear the level of specificity in the connections between L1-3 of V1 and RL. It is clear that there is a relationship for responses to high speeds along the horizontal axis, but these cells are responding to many other stimuli as well — so describing these connections as ‘highly-specific’ (line 285), seems not well supported.

We fully agree with this point, and in the revised manuscript we have changed this sentence so that it now reads (page 3):

“Our results indicate there is a cortical space for retinal direction selectivity and a distinct pathway that enables specialized response properties in HVAs.”

Furthermore, we removed “highly specific” from the Discussion section so that it now reads (page 9, line 24):

“Second, we identified a connection motif in the cortico-cortical V1 L2/3 projection neurons, by which feedforward signaling, originating from retinal DS cells, is selectively routed to area RL, but not to area PM (Fig. 7j).”

It would be helpful to readers to say more about the specificity of the *Frmd7* line in the Discussion. How confident are we that the perturbations in this line are retina-specific? Are there any caveats to the interpretation that all the effects observed in HVAs result from changes in the retina?

Thank you for this important comment. Regarding the specificity of *Frmd7* expression: as we have demonstrated previously, *Frmd7* expression is not observed in any of the major visual areas such as the dLGN or visual cortex (Yonehara et al., Neuron 2016). We have thus a high level of confidence that the perturbations in this line are retina specific in the context of our experiments. Furthermore, in the revised manuscript we report ablation experiments wherein we specifically ablated starburst amacrine cells by genetic targeting the diphtheria toxin receptor to ChAT-positive cells and intravitreally injected diphtheria toxin, yielding retina-specific and acute perturbations of retinal direction selectivity (see also Hillier et al., Nat Neurosci 2017 for the retinal specificity of this ablation). The results obtained from two-photon calcium imaging in areas RL and PM of these starburst-ablated mice were very similar to the results obtained in *Frmd7^m* mice (see Supplementary Figs. 3 and 5). These data support the conclusion that effects observed in HVAs are indeed due to specific impairments in retinal direction selectivity.

Reviewers' Comments:

Reviewer #1:

Remarks to the Author:

The authors responded to all my previous concerns and requests for new data and clarification of which cells are impacted by the mutation used.

It is a thorough, novel and important paper on DS visual circuits that will have important implications for the understanding of sensory processing, cortical local and sensory-long range circuits.

Reviewer #2:

Remarks to the Author:

The revised manuscript has been improved substantially. It addressed all the concerns I raised and will be an important contribution to the field, justifying the publication in Nature Communications. I recommend the study be published without further delay.

Reviewer #3:

Remarks to the Author:

The authors have successfully addressed my comments. I have one suggestion regarding color maps in the figures. The blue -> yellow color map works well for values that vary from 0 to positive values. However, many of the plots (e.g. Fig 2a,e,h,i, Fig 4a,e, Fig 5f,j) have color maps that represent values that range from negative to positive values, and whether those values are above or below zero is important. Thus, it would be a bit better to utilize a colormap that varies from e.g. red-to-blue and passes through white at 0. This would allow the reader to quickly tell whether the value is above or below zero.

Point-by-point list of responses to the reviewers' comments

We genuinely thank the three reviewers for their positive comments on our revised manuscript. Our comments are written below the original comments made by reviewers 1, 2, and 3.

REVIEWERS' COMMENTS:

Reviewer #1 (Remarks to the Author):

The authors responded to all my previous concerns and requests for new data and clarification of which cells are impacted by the mutation used. It is a thorough, novel and important paper on DS visual circuits that will have important implications for the understanding of sensory processing, cortical local and sensory-long range circuits.

We thank the reviewer for the positive attitude toward our manuscript and would once again like to express our gratitude for the insightful and constructive comments we received on our original manuscript.

Reviewer #2 (Remarks to the Author):

The revised manuscript has been improved substantially. It addressed all the concerns I raised and will be an important contribution to the field, justifying the publication in Nature Communications. I recommend the study be published without further delay.

We are happy to hear that the reviewer found the revised manuscript improved and worthy of publication in Nature Communications. We would like to take the opportunity to thank the reviewer for the valuable and important comments we receive on our original manuscript.

Reviewer #3 (Remarks to the Author):

The authors have successfully addressed my comments. I have one suggestion regarding color maps in the figures. The blue -> yellow color map works well for values that vary from 0 to positive values. However, many of the plots (e.g. Fig 2a,e,h,i, Fig 4a,e, Fig 5f,j) have color maps that represent values that range from negative to positive values, and whether those values are above or below zero is important. Thus, it would be a bit better to utilize a colormap that varies from e.g. red-to-blue and passes through white at 0. This would allow the reader to quickly tell whether the value is above or below zero.

We thank the reviewer for pointing out this important issue. We fully agree that the colormap used in the revised manuscript was not ideal for quickly interpreting the panels. Therefore, in the newest version of the manuscript, we have updated the colormaps in the relevant figures, such that it now utilizes a three-color colormap (i.e. blue-white-red). The updated panels for area RL from Figure 2 are shown here, together with its corresponding legend.

Fig. 2: Neurons with distinct functional characteristics are sensitive to disruption of retinal horizontal direction selectivity in area RL. (a) Response matrix composed of TF-dependent response amplitudes and DSI for RL L2/3 DS cells sorted by TF preference in control and *Frmd7tm* mice. (b) 2D visualization of the 1st and 2nd principal components for the response matrix shown in (a). Each point represents one neuron. (c) TF preference of individual RL neurons in control and *Frmd7tm* mice. (d) Fraction of neurons in 8×8 grids (gray lines) calculated from the PCA plot shown in (b) for control and *Frmd7tm* mice. (e) Fraction difference map between control and *Frmd7tm* mice. Black and white asterisks: significantly decreased and increased fractions in *Frmd7tm* mice, respectively, $P < 0.05$, χ^2 test with Yates correction.